# Qualitative Analysis and Componential Differences of Chemical Constituents in Taxilli Herba from Different Hosts by UFLC-Triple TOF-MS/MS

**DOI:** 10.3390/molecules26216373

**Published:** 2021-10-21

**Authors:** Jiahuan Yuan, Li Li, Zhichen Cai, Nan Wu, Cuihua Chen, Shengxin Yin, Shengjin Liu, Wenxin Wang, Yuqi Mei, Lifang Wei, Xunhong Liu, Lisi Zou, Haijie Chen

**Affiliations:** 1College of Pharmacy, Nanjing University of Chinese Medicine, Nanjing 210023, China; 20200655@njucm.edu.cn (J.Y.); caizhichen2008@126.com (Z.C.); wunan7272@163.com (N.W.); cuihuachen2013@163.com (C.C.); yinshengxin723@163.com (S.Y.); wangwenxin66666666@163.com (W.W.); 18260028173@163.com (Y.M.); weilifangquiet@163.com (L.W.); zlstcm@126.com (L.Z.); chenhaijie039X@163.com (H.C.); 2College of Pharmacy, Guangxi University of Chinese Medicine, Nanning 530220, China; 000682@gxtcmu.edu.cn

**Keywords:** Taxilli Herba, hosts, chemical constituents, UFLC-Triple TOF-MS/MS

## Abstract

Taxilli Herba (TH) is a well-known traditional Chinese medicine (TCM) with a wide range of clinical application. However, there is a lack of comprehensive research on its chemical composition in recent years. At the same time, *Taxillus chinensis* (DC) Danser is a semi parasitic plant with abundant hosts, and its chemical constituents varies due to hosts. In this study, the characterization of chemical constituents in TH was analyzed by ultra-fast liquid chromatography coupled with triple quadrupole-time of flight tandem mass spectrometry (UFLC-Triple TOF-MS/MS). Moreover, partial least squares discriminant analysis (PLS-DA) was applied to reveal the differential constituents in TH from different hosts based on the qualitative information of the chemical constituents. Results showed that 73 constituents in TH were identified or tentatively presumed, including flavonoids, phenolic acids and glycosides, and others; meanwhile, the fragmentation pathways of different types of compounds were preliminarily deduced by the fragmentation behavior of the major constituents. In addition, 23 differential characteristic constituents were screened based on variable importance in projection (VIP) and *p*-value. Among them, quercetin 3-*O*-*β*-D-glucuronide, quercitrin and hyperoside were common differential constituents. Our research will contribute to comprehensive evaluation and intrinsic quality control of TH, and provide a scientific basis for the variety identification of medicinal materials from different hosts.

## 1. Introduction

The traditional Chinese medicine Taxilli Herba (TH) is the dried stems and branches with leaves of *Taxillus chinensis* (DC.) Danser. It is a famous genuine medicinal material of Guangxi Province in China, with the properties of dispelling rheumatism, nourishing liver and kidney, strengthening muscles and bones, and miscarriage prevention. TH is frequently prescribed for rheumatic arthralgia, waist and knee weakness, muscle weakness, metrorrhagia, bleeding during pregnancy, fetal movement, dizziness, and other symptoms [1]. Modern pharmacological studies showed that TH has significant effects on anti-inflammatory and analgesic, anti-tumor, lowering blood pressure, lowering blood sugar, and protecting nerves and so on [2]. Chemical composition is the material basis of clinical efficacy. Phytochemical analysis has revealed that TH contains multiple chemical constituents such as flavonoids [3,4,5,6,7], phenolic acids [5], volatiles [8,9,10,11], terpenoid derivatives [12], and other chemical constituents based on previous literature. However, the chemical constituents of TH is still lack of in-depth analysis. Flavonoids were recommended as the inspection indicators in the quality evaluation reports, mainly focusing on the quantitative determination of quercetin, quercitrin and avicularin. Therefore, it is of great significance to clarify the main chemical constituents of TH for better control the quality of medicinal materials.

Since *Taxillus chinensis* (DC.) Danser is a semi-parasitic plant, the complex diversity of host plants constitutes an important biological feature of TH. According to the results of the resource survey, there are currently more than 150 kinds of hosts for TH. Nevertheless, it is difficult to distinguish TH from different hosts based on their appearance. Simultaneously, the host plants affect the quality of TH through the special relationship between the hosts and TH in terms of chemical constituents and pharmacological effects [13]. Hence, distinguishing the differences in the chemical constituents of TH from different hosts is also extremely necessary and important.

In recent years, Liquid chromatography-mass spectrometry (LC-MS) technique has become the most widely used analytical method for direct identification of multiple constituents in traditional Chinese medicine (TCM), because it combines the high separation performance of chromatography with the high discrimination ability of mass spectrometry. Among them, ultra-fast liquid chromatography coupled with triple quadrupole-time of flight tandem mass spectrometry (UFLC-Triple TOF-MS/MS) has complementary advantages, with strong separation ability, high detection sensitivity and strong specificity, etc [14]. Partial least squares discriminant analysis (PLS-DA) is a supervised statistical method of discriminant analysis, which can be used to establish a model of the relationship between the expression of metabolites and the sample category to realize the prediction of the sample category. At present, PLS-DA is widely used in the quality control of traditional Chinese medicines, such as the authentication identification of medicinal materials, the identification of base sources, and the rapid identification of medicinal materials of different origins [15,16,17,18]. Thus, in this study, qualitative analysis of TH from *Morus alba* L. was carried out based on UFLC-Triple TOF-MS/MS. A total of 73 constituents were identified by UFLC-Triple TOF-MS/MS and the fragmentation pathways of different types of compounds was summarized according to the fragmentation behavior of the major constituents. PLS-DA was applied to discriminate TH samples from seven common hosts based on the above qualitative results. 23 differential characteristic constituents were identified according to variable importance in projection (VIP) and *p*-value. Among them, quercetin 3-O-*β*-*D*-glucuronide, quercitrin and hyperoside were the common differential constituents. Our study could be conducive to the standard formulation and comprehensive quality control of TH and could also provide a scientific basis for the identification of TH from different hosts.

## 2. Results 

### 2.1. Optimization of Extraction Conditions

In order to optimize the extraction conditions, several factors were examined with different concentrations of extraction solvent (30%, 40%, 50%, 60%, 70%, 80% and 100% methanol); solid-liquid ratio (1:10, 1:20, 1:30, 1:40 and 1:50, *w/v*); and extraction time (15, 30, 45, 60, 75 and 90 min), which might have different effects on extraction efficiency. The results showed that the chromatogram had the most peaks and the extraction efficiency was relatively high with the conditions of a 1:30 ratio in 50% methanol for 30 min at room temperature.

### 2.2. Optimization of UFLC-Triple TOF-MS/MS Conditions

The effects of methanol-water, acetonitrile-water, methanol−0.4% (*v*/*v*) formic acid water solution, methanol: acetonitrile (1:1)−0.4% (*v*/*v*) formic acid water solution as the mobile phase, flow rates (0.8 and 1.0 mL/min), and column temperatures (25, 30, 35 °C) on the resolution of each peak in the samples were compared to achieve higher separation. The results showed that each peak could achieve a good separation effect when we chose methanol: acetonitrile (1:1)−0.4% (*v*/*v*) formic acid water solution as the mobile phase.

### 2.3. Identification of the Constituents in TH

The base peak chromatogram (BPC) of TH sample (S1–4, 4 batches of Taxilli Herba samples from *Morus alba* L. were numbered S1-1, S1-2, S1-3, S1-4.) in the negative ion mode was shown in Figure 1. Finally, 73 constituents were identified, including 33 flavonoids, 7 phenolic acids, 4 phenylpropanoids, 5 tannins, 13 glycosides, and 11 other constituents. Among them, 15 compounds were identified by comparison with the retention time and characteristic fragment ions of the standards, and the rest were speculated based on databases and related literature. The detailed information of the identified compounds was shown in Table 1, with their corresponding structures in Figure 2.

#### 2.3.1. Identification of Flavonoids

Flavonoids are the main active ingredients of TH. A total of 33 flavonoids were identified in this study, including dihydroflavones, dihydroflavonols, flavonols, isoflavones, flavones, flavanes, and other flavonoids.

In the structure of various flavonoids, the substituents on the A and B rings are mostly hydroxyl, methyl, and and methoxy groups, while the C ring is generally connected to monosaccharides or polysaccharides. The basic cleavage methods are loss of neutral fragments and the Retro-Diels-Alder (RDA) cleavage of the C ring. Several RDA cleavage modes of flavonoids were shown in Figure 3.

Dihydroflavones and dihydroflavonols: compounds **30** and **71** were identified as dihydroflavones, and compounds **41**, **55**, and **64** were identified as dihydroflavonols. It can be seen from the fragment ions of these compounds that dihydroflavonoids generally do not lose neutral fragments such as CO (28 Da) and CO_2_ (44 Da). Dihydroflavonoids are prone to have RDA reactions, where 1, 3 bonds of C ring are more likely to break to produce [^1,3^A]^−^ and [^1,3^B]^−^. Compound **71** (Figure 4A) was substituted by glucose at position 7 in A ring, and it was speculated that there were two possible cleavage pathways inferred based on the MS/MS spectrum. The first pathway was to break the 1,3 bonds of the C ring directly, producing a fragment with a sugar group, the second pathway was to lose glycosides to obtain aglycones, and then break the 1, 3 bonds of C ring. The cracking law of dihydroflavonols is similar to that of dihydroflavones. Although compounds **55** and **64** had the same glucose groups in their structures, the substitution positions were different. Compound **55** lost one molecule of glycoside and then RDA reaction occurred, while the fragment ions generated by compound **64** were different from that of compound **55**. It followed that the position of the substituent had a great influence on the cleavage sequence of the sugar chain and the C ring.

Flavanes: compounds **34**, **35**, **39**, and **54** were flavan-3-ols as well as belonged to catechin compounds. The cleavage process generally occurred in the A, B, and C rings. Taking (+)-catechin as an example, Figure 4B showed an accurate mass of [M−H]^−^ ion at *m*/*z* 289.0724, which corresponded to the molecular formula of this compound as C_15_H_14_O_6_. The compound was identified as (+)-catechin based on the mass spectrometry data in literature and comparied with the reference substance. There were numerous breaks between 1 and 2, 3, 4 bonds in the C ring resulting in 167.0339 [^1,2^A]^−^, 137.0234 [^1,3^A]^−^, 125.0235 [^1,4^A]^−^. Successive loss of H_2_O and B ring (C_6_H_6_O_2,_ 109 Da) generated the fragment ion at *m*/*z* 163.0385, and the fragment ion at *m*/*z* 179.0341 was produced by losing the B ring directly without losing water. The ion at *m*/*z* 245.0235 suggested that the A ring lost a neutral fragment of CO_2_. The others had similar cracking laws.

Flavonoids existed as glycosides or completely in free form in plants. Compounds **19**, **42**, **66**, **50**, and **70** were classified as flavones. In the primary mass spectrum, the flavones all showed the quasi-molecular ion peak [M−H]^−^ without other fragment ions. Flavones exhibited some of the same cracking patterns in MS^2^, such as the loss of neutral fragments of CO, CO_2_, H_2_O (18 Da), CH_3_ (15 Da), and OCH_3_ (31 Da). In the negative ion mode, the C ring broke to produce ^1,3^A^−^ (151 Da), ^1,3^ B^−^ (133 Da), and (^1,3^A-CO_2_)^−^ (107 Da), of which ^1,3^A^−^ was the main fragment ion. The fragment ions generated in the second mass spectrum would also increase in parallel with the number of hydroxyl substitutions increasing. Compounds 19 and 66 used luteolin as the basic nucleus, with characteristic fragment ions at *m*/*z* 285 and *m*/*z* 151 (^1,3^A^−^) in the second mass spectrum. Neither compound lost fragments of C_3_O_2_, which might be related to the substituents on the A ring. Compounds **19**, **42**, **50**, and **66** had hydroxyl substitutions at different positions on the B ring. The compounds with substituents on 3’, 4’, and 5’ would not break between C1 and C4. The compound **70** only lost OCH_3_, CH_3_, and other fragments without other fragments of RDA reaction in the MS spectrum for the reason that the only substitution of OCH_3_ on the structure hid the cleavage of the C ring. Figure 4C showed the possible cleavage pathway of Luteolin-7-*O*-glucoside.

Flavonols: In the negative ion mode, flavonols usually experienced the loss of OH, CO, CO_2_, C_2_H_2_O, B ring as well as the cracking of the C ring. Compared with flavones, the C ring of flavonols was easier to open. For example, compound **73** (Figure 4D) showed that [M−H]^−^ molecular ion was at *m*/*z* 301.0354 and other abundant fragment ions, such as ions at *m*/*z* 273.0376, 178.9988 [^1,2^A]^−^, 151.0030 [^1,3^A]^−^, 121.0296 [^1,2^B]^−^, 107.0143 [^0,4^A]^−^. Among them, fragments were more common formed by the cracking of 1, 3 bonds. Meanwhile, it was compared with the standard product information and confirmed as quercetin. The structure of flavonol glycosides contained more hydroxyl groups, which easily formed [M−H]^−^ quasi-molecular ion peaks, and further removed the sugar chain to form aglycone (Y_0_^−^). Compounds **49**, **51**, **52**, **56**, **57**, **59**, **60**, **61**, **62**, **63**, **65**, **67**, **69**, and **72** were identified as flavonolosides. Among them, compounds **49**, **65**, **69**, and **72** were flavonol glycosides with kaempferol as the core, while **56**, **57**, **59**, **60**, **61**, **62**, **63**, and **67** were flavonol glycosides with quercetin as the core. Taking quercitrin as an example, the quasi-molecular ion peak of 447.0921 [M−H]^−^ was first formed. Fragment ion at *m*/*z* 301.0354, 283.0223 represented the neutral loss of rhamnose and H_2_O. The ion at 151.0024 was produced by the breakage of 1, 3 bonds. In the flavonoid glycosides with quercetin as the basic nucleus, characteristic ion could be seen as at *m*/*z* 301 after the loss of the sugar chain, which could be used as a basis for determining whether the core is quercetin. Similarly, with kaempferol as the basic nucleus, the characteristic fragment ions at *m*/*z* 285 could also be regarded as a basis to judge whether kaempferol is the core.

Other flavonoids: compounds **14** and **15** were identified as bisphenirone flavonoids, which were a special type of flavonoids with a C6-C1-C6 skeleton. Compound 21 was identified as a flavonoid lignan compound with a complex structure, and compound 68 was identified as isoflavones.

#### 2.3.2. Identification of Phenolic Acids

The mass spectrometry cleavage behavior of phenolic acids was relatively simple. In the negative ion mode, the primary mass spectrum mainly existed in the form of molecular ion peaks of [M−H]^−^. The secondary mass spectrum mainly showed the loss of CO_2_ and H_2_O resulting in [M−H−CO_2_]^−^ or [M−H−H_2_O]^−^ fragment ions. Compounds **5**, **6**, **7**, **8**, **9**, **23**, and **32** were identified as phenolic acid. Fragments after losing CO_2_ or COOH (45 Da) were usually seen in the mass spectrogram due to the common feature inclusion of COOH groups in these compound structures. Loss of substituents also occurred if the compound had other substituents such as hydroxyl groups. For example, compound **9** was speculated that its molecular formula might be C_7_H_6_O_5_ based on the ion [M−H]^−^ at *m*/*z* 169.0138. The fragment ions at 125.0240, 107.0141, 97.0341, and 69.0374 were inferred to be caused by the loss of CO_2_, H_2_O, and CO. Finally, it was verified that the compound corresponding to peak 9 was gallic acid (Figure 4E). In the same way, compounds **5, 6**, **7**, **8**, **23**, **32** were speculated as quinine acid, shikimic acid, malic acid, citric acid, protocatechuic acid, and 4-hydroxybenzoic acid, respectively. Fragment ions after loss of CO_2_ and COOH were shown in the MS^2^ of these 7 compounds. With different amounts of hydroxyl substitutions in compounds **6**, **7**, **9**, **23**, and **32**, varying degrees of losing H_2_O could be seen in the corresponding fragments. The fragment information was shown in Table 2.

#### 2.3.3. Identification of Phenylpropanoids

Phenylpropanoids were generally a class of compounds composed of C6-C3 as the basic unit, including simple phenylpropanoids, coumarins, and lignans. Our study inferred 4 phenylpropanoids, including simple phenylpropanoids (**25**, **36**, **38**) and coumarins (**17**), respectively. Simple amphetamine compounds were more likely to lose neutral molecules such as H_2_O, CO, and CO_2_ during the cracking progress. For example, compound **36** continuously lost H_2_O, CO, and CO_2_ resulting in the generation of fragment ions such as [M−H−CO_2_−CO]^−^ (109.0440 Da), [M−H−CO_2_−CO−H_2_O]^−^ (89.0413 Da), The compound 36 was identified ultimately as caffeic acid based on the primary mass spectrometry and secondary debris ions of the compound combined with relevant literature. Similarly, fragment ions after loss of CO_2_ were presented in MS^2^ of p-Coumaric acid. Compound **38** chlorogenic acid belonged to the class of caffeoylquinic acid, and there were two possible cleavage methods at the position of the ester bond. One was that the acyl-oxygen bond fractured leading to the loss of a molecule of caffeoyl (162 Da) and obtained an ion at *m*/*z* 191.0554. The second was that the fragments ions at at *m*/*z* 173 and 179 were obtained after the alkoxy group were broken. The possible cleavage pathway was shown in Figure 4F.

#### 2.3.4. Identification of Tannins

Compounds **11**, **26**, **27**, **28**, and **45** belonged to tannins, among which **11** were hydrolysable tannins and the rest were condensed tannins. The basic composition of condensed tannins is catechin/epicatechin, which is a polymer formed by polymerization of C4-C6 bonds or C4-C8 bonds (esters formed by dehydrated with gallic acid). The cracking methods of proanthocyanidin polymers mainly included the fragmentation between flavanes and the RDA reaction. There were two possibilities for the break between flavanes. On the one hand, it lost the neutral fragments of the top unit T-unit (TOP) which was only connected to other units by C4 bonds to form the fragment at *m*/*z* 287. On the other hand, it lost the bottom unit B-unit (BASE) which was connected to other units by C6 or C8 bonds to form the fragment at *m*/*z* 289. Flavanes generally underwent a RDA reaction and lost a neutral structure of C_8_H_8_O_3_ (152 Da). Taking the procyanidin B2 (Figure 4G) as an example, the ion fragment at *m*/*z* 289 was produced by losing a molecule of catechin. The fragment ion at *m*/*z* 425 was produced by RDA rearrangement, and an ion at *m*/*z* 407 was produced by continuing to lose one molecule of H_2_O based on the ion fragment at *m*/*z* 425. Another possible way of cleavage was to lose one molecule of H_2_O first, and then lose one C_6_H_5_O_2_ (109 Da) fragment to get an ion at *m*/*z* 451. The cleavage law of trimer procyanidin C1 and proanthocyanidin B2 was virtually identical. At the same time, there was also a dimer anion fragment at *m*/*z* 577 by losing one top-unit and a monomeric anion fragment at *m/z* 289 by losing two top-units. Compound 45 showed a [M−H]^−^ ion at *m*/*z* 729.1441. The fragment ions at *m*/*z* 577.1324 and 441.0743 were produced by the loss of C_7_H_4_O_4_ (152 Da) and a T-unit. The specific fragment information was shown in Table 2.

#### 2.3.5. Identification of Glycosides

Glycosides were a class of compounds formed by connecting saccharides or saccharides derivatives with another non-sugar substance through the carbon atom of the terminal group of the sugar. Compounds **12**, **13**, **18**, **24**, **31**, **33**, **37**, **40**, **43**, **46**, **47**, **48**, and **58** were identified as glycosides. The 13 compounds were all oxyglycosides formed by connecting oxygen atoms with sugars, among which compounds **33** and **58** were ester glycosides and the others were phenol glycosides, respectively. The characteristic ion fragments after the loss of one glucose (162 Da) could be seen clearly from the MS^2^ of these compounds. The ion at *m*/*z* 101.0437 was formed when compound **43** lost one molecule of xylose and glucose successively. Compound **24** was linked to glucuronic acid, and the fragment ion after the loss of glucuronic acid (176 Da) could also be clearly visible from MS^2^. The details were shown in Table 2.

### 2.4. Analysis of the Differential Constituents of TH from Different Hosts

#### 2.4.1. PLS-DA of the Samples

A pattern-supervised identification method PLS-DA analysis was used to compare the chemical constituents in TH from different hosts comprehensively. The potential differential chemical constituents were found based on the VIP obtained from the PLS-DA model, and the T-test was used to verify whether the differential chemical constituents in multi-dimensional statistics had significant differences in one-dimensional statistics, where *p* < 0.05 indicated significant differences. In this experiment, the samples from the other 6 common hosts were compared with the samples from *Morus alba* L. and analyzed by PLS-DA. The results were shown in Figure 5. Two samples from different hosts were clearly separated along the PIC axis, and the model verification results (R^2^Y = 0.496, 0.123, 0.602, 0.034, 0.001, 0.153; Q^2^ = –0.207, –0.247, –0.297, –0.264, –0.263, –0.289, respectively.) showed that the models were effective and reliable.

#### 2.4.2. Identification of the Differential Chemical Constituents

A total of 23 differential chemical constituents were initially identified in samples from 7 hosts, including monotropein, procyanidin B2, procyanidin B2 3’-*O*-gallate, hyperoside, quercetin 3-*O*-*β*-*D*-glucuronide, isoquercitrin, quercitrin, procyanidin B1, procyanidin C1, (+)-catechin, (−)-epicatechin gallate, acteoside, narcissin, (−)-epiafzelechin 3-*O*-gallate, cascaroside A, apigenin-7-*O*-rutinoside, apimaysin, kaempferitrin, isohemiphloin, isoastilbin, astilbin, rhamnitrin, neohesperidin dihydrochalcone. The results and the 85 constituents identified in different samples from 7 hosts are shown in Table 2 and Appendix A, respectively 2.4.3. Relative Content of Common Differential Chemical Constituents.

The three common differential constituents were quercetin 3-*O*-*β*-*D*-glucuronide, quercitrin and hyperoside. The relative content was represented by the corresponding peak area of common differential constituents in each group of samples. The average value and standard deviation of the peak area of the same chemical constituent in different samples were calculated to obtain the relative content changes of common different constituents between different samples. The results showed that TH from *Morus alba* L. contained higher levels of these 3 constituents, and TH from *Ilex latifolia* Thunb. contained high relative content of quercetin 3-*O*-*β*-D-glucuronide and quercitrin, and TH from *Passiflora edulia* Sims. contained high relative content of quercitrin and hyperoside. The results were shown in Figure 6.

## 3. Discussion

As mentioned previously, Taxilli Herba is a semi parasitic plant with complex hosts. The demand for TH in clinical is gradually increasing as well. In recent years, there have been few research reports on the chemical composition of TH. What’s more, the TH from different hosts currently circulating on the market are difficult to distinguish based on their appearance. In this study, we tried to establish a methodology to exploring the chemical constituents in TH. There were 73 chemical constituents identified ultimately in TH from *Morus alba* L., and flavonoids were the main constituents (Table 1). The scores scatter plot of PLS-DA showed that the samples from *Morus alba* L. and other hosts were significantly divided into two groups (Figure 5). 23 differential chemical constituents were initially identified of samples from 7 hosts, and the relative contents of three common differential constituents of quercetin 3-*O*-*β*-*D*-glucuronide, quercitrin and hyperoside in TH from *Morus alba* L. were higher than that of samples from other hosts (Figure 6). The results revealing possible components in TH will help us to have a deeper understanding of this medicine material, and can also be used as a basis for distinguishing samples of TH from different hosts. As far as the current situation is concerned, the diversified sources of medicinal materials are an important reason for the uneven quality of TH. At present, there are many medicinal materials from different host plants on the market, and TH from *Morus alba* L. is the most widely used clinically. However, the impact of the hosts on the quality of the medicinal materials in many aspects is still unknown. Systematic research on multiple levels from ingredients to curative effects to explain whether the effects of TH from different hosts are the same or different is also a question worthy of discussion. The most important thing is that this study could provide basic information for the quality formation of TH.

## 4. Materials and Methods

### 4.1. Chemicals and Reagents

The standard substances of isosakuranetin and quercetin 3-*O*-*β*-*D*-glucuronide were purchased from Nanjing Liangwei Biotechnology Co., Ltd. (Nanjing, China). Hyperin, auicularin, catechin, quercetin 3-*O*-(6″-*O*-galloyl)-*β*-galactoside and quercetin 3-*O*-(6″-*O*-galloyl)-*β*-glucoside were supplied by Shanghai Yuanye Biotechnology Co., Ltd. (Shanghai, China). Chlorogenic acid was received from Baoji Chenguang Biotechnology Co., Ltd. (Baoji, China). Isolquercitrin was provided by Chengdu Chroma Biotechnology Co., Ltd. (Chengdu, China). Kaempferitrin was obtained from Chengdu Alfa Biotechnology Co., Ltd. (Chengdu, China). Protocatechuic acid was acquired from Shanghai Winherb Medical Technology Co., Ltd.(Shanghai, China). Quercetrin was purchased from the National Institute for the control of Pharmaceutical and Biological Products (Beijing, China). Rutin, quercetin, and gallic acid were purchased from the National Institutes for Food and Drug Control (Beijing, China). The purity of all compounds was more than 98% determined by HPLC. Formic acid, methanol, and acetonitrile of HPLC grade (Merck, Darmstadt, Germany). Ultra-pure water was prepared by a Milli-Q water purification system (Millipore, Bedford, MA, USA).

### 4.2. Plant Materials

TH from 7 different hosts were collected from two regions in Guangxi Provice in China, and 4 batches of samples from each host were dried under the same conditions. See Table 3 for detailed information. The botanical origins of the materials were authenticated by Professor Xunhong Liu (Department for Authentication of Chinese Medicines, School of Pharmacy, Nanjing University of Chinese Medicine, Nanjing, China). Voucher specimens were deposited in the laboratory of Chinese medicine identification, Nanjing University of Chinese Medicine.

### 4.3. UFLC-Triple TOF-MS/MS Analysis of TH

#### 4.3.1. Preparation of Standard and Sample Solutions

A mixed standard stock solution of 15 standard substances (quercetin3-*O*-*β*-*D*-glucuronide, isosakuranetin, quercetin 3-*O*-(6″-*O*-galloyl)-*β*-galactoside, quercetin 3-*O*-(6″-*O*-galloyl)-*β*-glucoside, hyperin, auicularin, isolquercitrin, chlorogenic acid, catechin, kaempferitrin, protocatechuic acid, quercetrin, rutin, quercetin, and gallic acid) was prepared with 50% methanol at a final concentration of 5 μg/mL. The diluted solutions were used for UFLC-Triple TOF MS/MS analysis. All the solutions were stored at 4 °C for further analysis.

All samples were crushed and passed through 50-mesh. Accurately 0.5 g of TH powders were weighed and ultrasonically extracted with 15 mL of 50% methanol for 30 min in a conical flask at room temperature. After the extraction was paused for a few minutes, the supernatant was taken and centrifuged at 13,000 rpm/min for 10min (H1650-W high speed centrifuge, Hunan Xiangyi Laboratory Instrument Development Co., Ltd., Hunan, China). The supernatant was filtered through 0.45 μm membrane (Jinteng laboratory equipment Co., Ltd., Tianjin, China) prior to injection of UFLC-Triple TOF-MS/MS analysis.

#### 4.3.2. UFLC-Triple TOF-MS/MS Conditions

The UFLC system (Shimadzu., Kyoto, Japan) was used for sample analysis. The separation was conducted by an Agilent ZORBAX SB-C18 column (4.6 mm × 250 mm, 5 μm) at 30 °C. The mobile phase was composed of methanol-acetonitrile (1:1) (A) and 0.4% formic acid water (B) with the gradient elution: 0–5 min, 2–6% A; 5–6 min, 6–10% A; 6–8 min, 10–15% A; 8–12 min, 15–18% A; 12–18 min, 18–21% A; 18–21min, 21–23% A; 21–26 min, 23–25% A; 26–30 min, 25–27% A; 30–33 min, 27–40% A; 33–38 min, 40–50% A; 38–40 min, 50–2% A; 40–45 min, 2–2% A. The injection volume was 10 μL and the flow rate was 1.0 mL/min.

Mass spectrometry (MS) detection was performed by AB Sciex Triple TOF TM 5600 system-MS/MS (AB SCIEX, Framingham, MA, USA), equipped with an electrospray ionization (ESI) source in negative ion mode. The optimized MS conditions were set as follows: the ion source temperature (TEM): 600 °C; the flow rate of curtain gas (CUR): 40 psi; the flow rate of nebulization gas (GS1): 60 psi; the flow rate of auxiliary gas (GS2): 60 psi; the ion spray voltage floating (ISVF): 4500 V; the collision energy: −10 V; the declustering potential: −100 V. TOF MS and TOF MS/MS were scanned with the mass range of *m*/*z* 100~2000 and 50~1500, respectively.

#### 4.3.3. Identification of the Constituents

On the one hand, it was identified by compared with the previously established chemical composition database, and verified with the retention time and mass spectrometry data of the standards. On the other hand, the identification of other unknown chemical composition was inferred based on the fragment information of MS/MS with the combination of SciFinder (https://scifinder.cas.org/), HMDB (https://hmdb.ca/), CNKI (https://kns.cnki.net/) and related literature.

### 4.4. Analysis of Differential Constituents in TH from Different Hosts

#### 4.4.1. Chromatographic Processing and Statistical Analysis

Mass spectrometry data processed by Peakview 1.2 (Sciex AB, Framinghan, MA, USA) and Markerview 1.2.1 (Sciex AB, Framinghan, MA, USA) software were imported into SIMCA-P 13.0 (Umetrics AB, Umea, Sweden) software for analysis. Based on the above qualitative results, PLS-DA using the SIMCA-P 13.0 sotfware (Umetrics AB, Umea, Sweden) was used to perform dimensionality reduction analysis on the data to obtain information about differences between groups. The difference chemical components of TH from different hosts were found according to the VIP and *p*-value obtained by the PLS-DA model.

#### 4.4.2. Identification of the Differential Chemical Constituents

Except for comparison with the constituents in Table 1, other unknown differential constituents were identified through databases and literature including SciFinder (https://scifinder.cas.org/), HMDB (https://hmdb.ca/), CNKI (https://kns.cnki.net/).

## 5. Conclusions

In our study, an efficient method based on UFLC-Triple TOF-MS/MS was established for the qualitative characterization of Taxili Herba from *Morus alba* L. The results showed that 73 constituents were identified in total including flavonoids and phenolic acids, etc. The fragmentation pathways of flavonoids, phenolic acids, phenylpropanoids, tannins and glycosides were preliminarily deduced by the fragmentation behavior of the major constituents. Simultaneously, the results of PLS-DA showed that TH samples from *Morus alba* L and other hosts were clearly separated. 23 differential characteristic constituents were screened based on PLS-DA scores plot and VIP plot, and three common differential constituents showed different changing laws. In a word, the results could help us have a clearer understanding of the chemical constituents of TH and reveal differential constituents in TH from different hosts. The findings will contribute to comprehensive evaluation and intrinsic quality control of TH and provide a scientific basis for the identification of TH from different hosts.

## Figures and Tables

**Figure 1 molecules-26-06373-f001:**
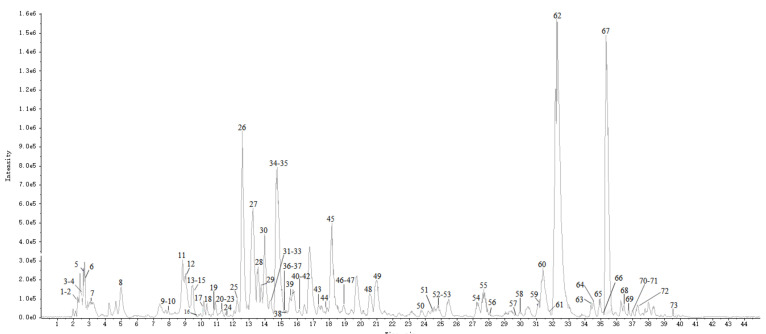
The base peak chromatogram (BPC) of Taxilli Herba from *Morus alba* L. in negative ion mode.

**Figure 2 molecules-26-06373-f002:**
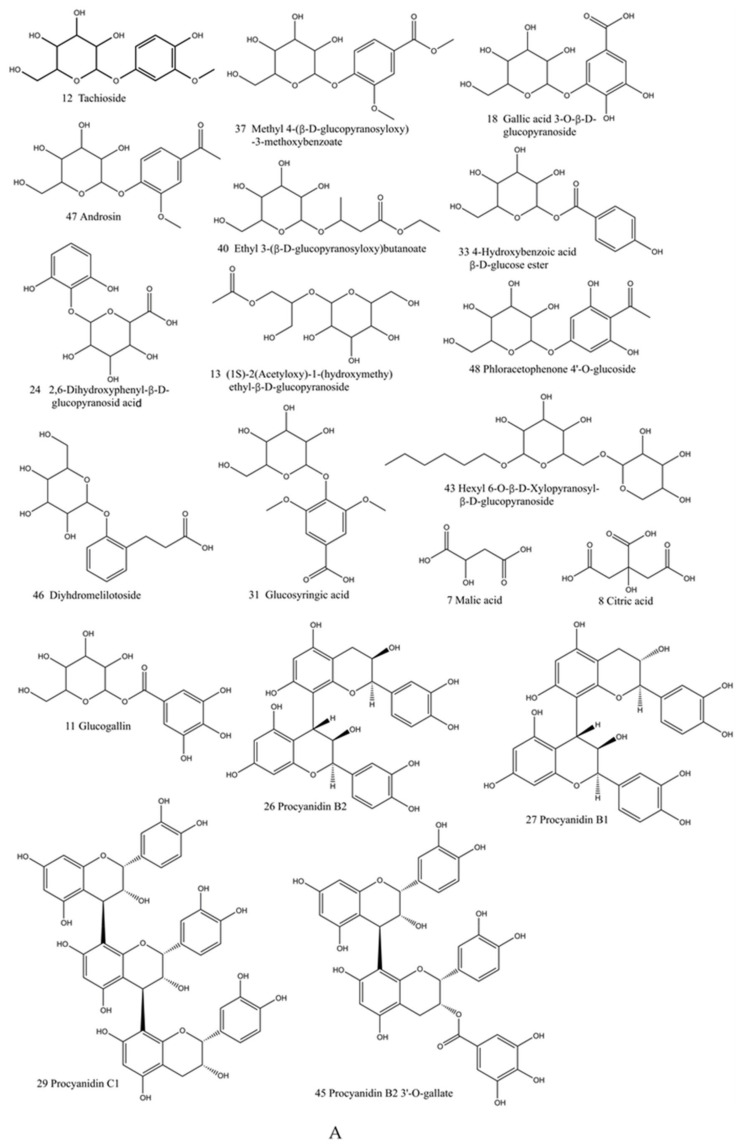
Chemical structures of compounds identified in the Taxilli Herba (**A**) exact structures, (**B**) general structures.

**Figure 3 molecules-26-06373-f003:**
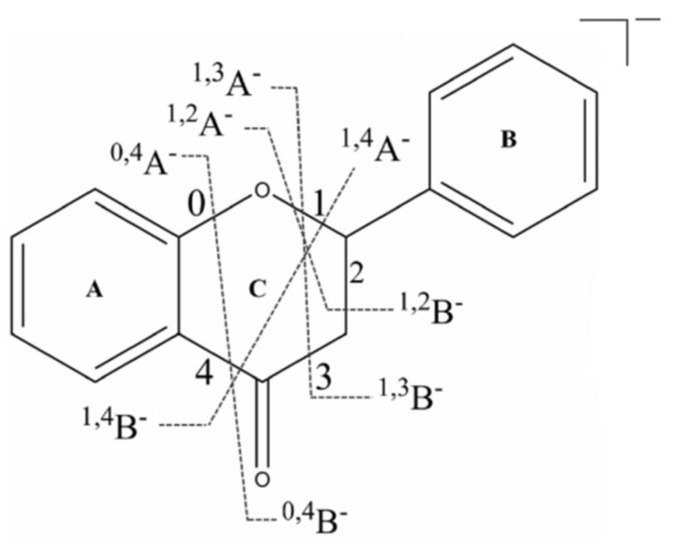
Schematic diagram of the fracture site of flavonoid aglycone in negative ion mode.

**Figure 4 molecules-26-06373-f004:**
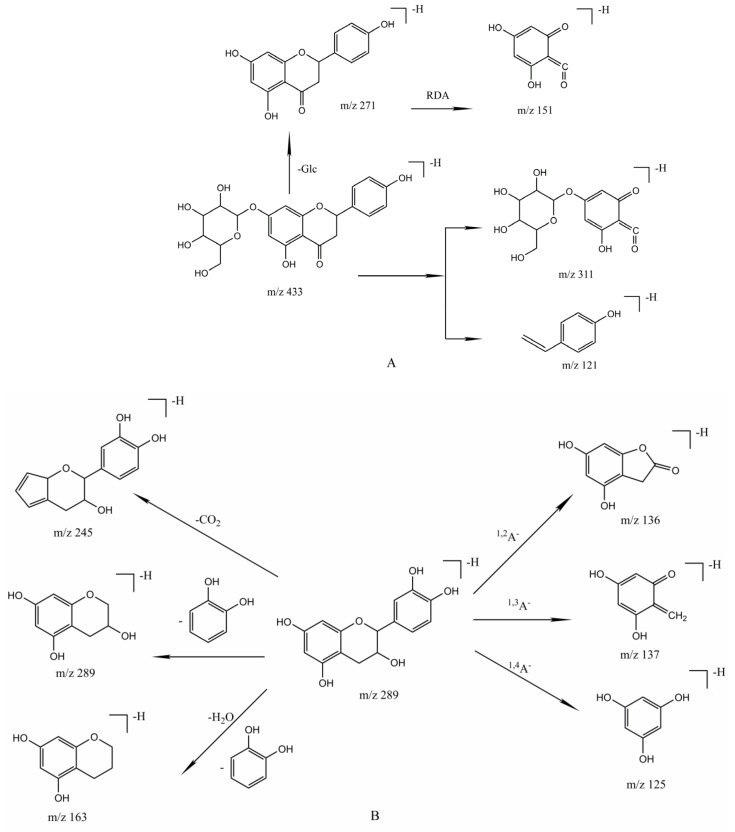
The possible fragmentation pathways of Prunin (**A**), (+)-catechin (**B**), Luteolin-7-O-glucoside (**C**), Quercetin (**D**), Gallic acid (**E**), Chlorogenic acid (**F**), and Procyanidin B2 (**G**) in Taxilli Herba.

**Figure 5 molecules-26-06373-f005:**
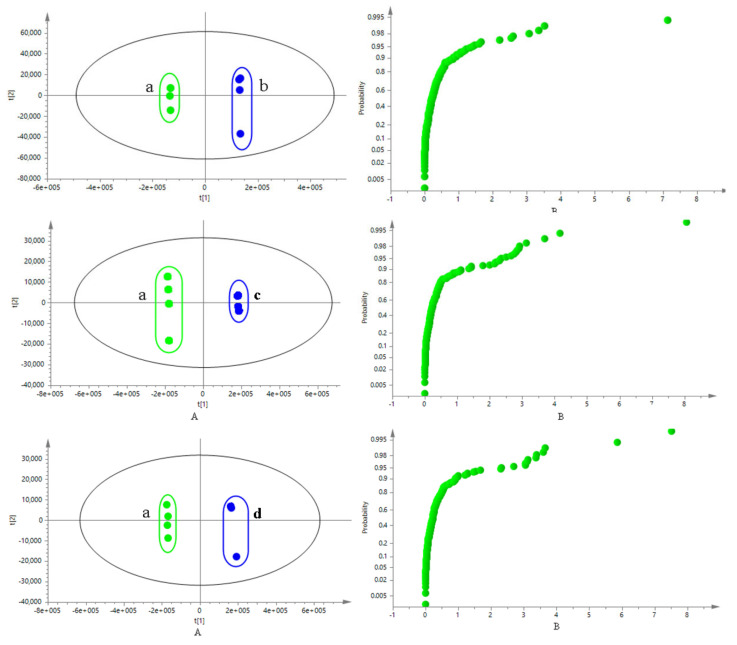
PLS-DA scores plot and VIP score plot of TH samples from different hosts. (*Morus alba* L. (**a**), *Liquidambar formosana* Hance. (**b**), *Ilex latifolia* Thunb. (**c**), *Crataegus pinnatifida Bge. var. major* N.E.Br. (**d**), *Passiflora edulis* Sims. (**e**), *Pyrus pyrifolia* (Burm. F.) Nakai. (**f**), and *Cinnamomum camphora* (L.) Presl (**g**)).

**Figure 6 molecules-26-06373-f006:**
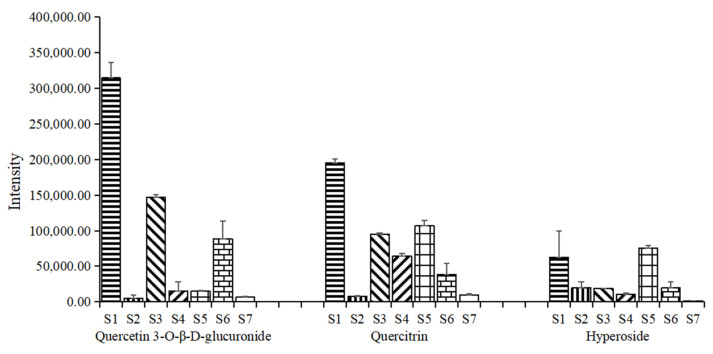
Relative contents of the common differential chemical constituents.

**Table 1 molecules-26-06373-t001:** Identification of 73 constituents in Taxilli Herba from *Morus alba* L. by UFLC-Triple TOF-MS/MS.

No.	t_R_ /min	Molecular Formula	MS^1^(*m/z*)	MS^2^(*m/z*)	Error /ppm	Compound	References
1	2.24	C_6_H_12_O_6_	179.0561[M−H]^−^	161.0457[M−H−H_2_O]^−^, 117.0207, 89.0248, 71.0158	0.05	Glucose	[19]
2	2.27	C_5_H_10_N_2_O_3_	145.0633[M−H]^−^	128.0539[M−H-NH_3_]^−^, 127.0514[M−H-H_2_O]^−^, 101.0725[M−H−CO_2_]^−^, 99.0560[M−H-HCOOH]^−^	0.70	Glutamine	[20]
3	2.54	C_11_H_20_O_10_	311.0991[M−H]^−^	233.0654[M−H−C_2_H_3_O_2_−CHOH]^−^, 173.0446[M−H−C_2_H_3_O_2_−CHOH−C_2_H_4_O_2_] ^−^,131.0341[M−H−C_6_H_11_O_6_]^−^, 99.0095[M−H−Xyl−C_3_H_6_O_2_]^−^, 71.0155[M−H−Xyl−C_4_H_8_O_3_]^−^	1.93	Primeverose	[21]
4 *	2.59	C_10_H_13_N_5_O_5_	282.0830[M−H]^−^	150.0426[M−H−Rib]^−^, 133.0153[M−H−Rib−H_2_O]^−^	4.90	Guanosine	[22]
5	2.63	C_7_H_12_O_6_	191.0057[M−H]^−^	173.0454[M−H−H_2_O]^−^, 127.0397[M−H−H_2_O−HCOOH]^−^, 59.0160, 71.0161, 85.0304	0.05	Quinic acid	[23]
6	2.86	C_7_H_10_O_5_	173.0472[M−H]^−^	155.0407[M−H_2_O]^−^, 137.0246[M−H−2H_2_O]^−^, 129.0189[M−H−CO_2_]^−^, 111.0451[M−H−CO_2_−H_2_O]^−^	1.10	Shikimic acid	[24]
7	3.12	C_4_H_6_O_5_	133.0146[M−H]^−^	115.0041[M−H−H_2_O]^−^, 71.0160[M−−H−H_2_O−CO_2_]^−^	2.60	Malic acid	[25]
8	4.96	C_6_H_8_O_7_	191.0206[M−H]^−^	173.0101[M−H−H_2_O]^−^, 154.9982[M−H−2H_2_O]^−^, 129.0187[M−H−H_2_O−CO_2_]^−^, 103.0400[M−H-2CO_2_]^-^	0.00	Citric acid	[26]
9 *	7.93	C_7_H_6_O_5_	169.0138[M−H]^−^	125.0240[M−H−CO_2_]^−^, 107.0141[M−H−CO_2_−H_2_O]^−^, 97.0341[M−H−CO2−CO]^−^, 69.0374[M−H−CO_2_−2CO]^−^	2.60	Gallic acid	[27,28]
10	8.00	C_19_H_16_O_4_	307.1029[M−H]^−^	145.0508, 127.0397	2.60	Bisdemethoxycurcumin	[29]
11	8.86	C_13_H_16_O_10_	331.0673[M−H]^−^	179.0137[M−H−G]^−^, 169.0253, 161.024[M−H−G−H_2_O]^−^	1.80	Glucogallin	[28]
12	9.03	C_13_H_18_O_8_	301.0928[M−H]^−^	139.0324[M−H−Glc]^−^, 123.0089[M−H−Glc−O]^−^	1.66	Tachioside	[26]
13	9.34	C_11_H_20_O_9_	295.1045[M−H]^−^	131.0934[M−H−Glc]^−^, 113.0252[M−H−Glc−H_2_O]^−^, 85.0306, 71.016, 59.0162	0.30	(1S)-2(Acetyloxy)-1-(hydroxymethy)ethyl-*β*-*D*-glucopyranoside	[30]
14	9.39	C_20_H_20_O_11_	435.1129[M−H]^−^	271.0448[M−H−Glc]^−^, 313.0354, 151.0037, 125.0245	2.52	Homomangiferin	[31]
15	9.43	C_24_H_20_O_8_	435.1124[M−H]^−^	313.0509, 151.0025, 123.0085	0.23	Isochinomin	[32]
16	9.55	C_26_H_22_O_10_	493.1183[M−H]^−^	331.0654[M−H−Gal]^−^	1.42	Fluorescein-*β*-*D*-galactopyranoside	[29]
17	10.16	C_23_H_18_O_7_	405.1031[M−H]^−^	169.0134, 71.0154	4.93	Toddacoumaquinone	[32]
18	10.31	C_13_H_16_O_10_	331.1061[M−H]^−^	169.0134[M−H−Glc]^−^, 125.0240[M−H−Glc−CO_2_]^−^	0.60	Gallic acid 3-*O*-*β*-*D*- glucopyranoside	[33,34,35]
19	10.77	C_22_H_16_O_6_	375.0927[M−H]^−^	285.0437[M−H−Benzyl group]^−^, 151.0030[^1,3^A]^−^	1.06	7-*O*-Benzyl Luteolin	[27]
20	11.12	C_24_H_20_O_7_	419.1177[M−H]^−^	271.0421, 151.0033	3.50	Artonol B	[36]
21	11.13	C_25_H_20_O_9_	463.0869[M−H]^−^	435.0761[M−H−CO]^−^, 273.0364[M−H−C_10_H_11_O_3_]^−^, 151.0927[^1,3^A]^−^	0.00	Hydrocarpin	[37,38]
22	11.30	C_26_H_30_O_14_	565.1547[M−H]^−^	403.0473[M−H−Glc]^−^, 241.0241[M−H−2Glc]^−^	2.80	Mulberroside F	[39]
23 *	11.38	C_7_H_6_O_4_	153.0194[M−H]^−^	109.0303[M−H−CO_2_]^−^, 101.0314[M−H−CO_2_−CO]^−^	6.90	Protocatechuic acid	[27]
24	11.68	C_12_H_14_O_9_	301.0565[M−H]^−^	283.0456[M−H−H_2_O]^−^, 125.0241[M−H−Glc UA]^−^, 107.0144[M−H−Glc UA−H_2_O]^−^	0.00	2,6-Dihydroxyphenyl-*β*-*D*- glucopyranosiduronic acid	[40]
25	12.24	C_9_H_8_O_3_	163.0401[M−H]^−^	119.0482[M−H−CO_2_]^−^, 93.0316[M−H−CO_2_−C_2_H_2_]^−^	0.18	p-Coumaric acid	[23]
26	12.57	C_30_H_26_O_12_	577.1354[M−H]^−^	451.0989[M−H−H_2_O−C_6_H_5_O_2_]^−^, 425.0835[M−H−C_8_H_8_O_3_]^−^, 407.0733[M−H−C_8_H_8_O_3_−H_2_O]^−^, 289.0685[M−H−TOP]^−^, 245.0768[M−H−TOP−CO_2_]^−^, 179.0733[M−H−TOP−C_6_H_5_O_2_]^−^, 125.0231[^1,4^A]^−^	0.40	Procyanidin B2	[41,42,43]
27	13.22	C_30_H_26_O_12_	577.1354[M−H]^−^	451.0986[M−H−H_2_O−C_6_H_5_O_2_]^−^, 425.0829[M−H−C_8_H_8_O_3_]^−^, 407.0745[M−H−C_8_H_8_O_3_−H_2_O]^−^, 289.0698[M−H−TOP]^−^, 245.0743[M−H−TOP−CO_2_]^−^, 179.0721[M−H−TOP−C_6_H_5_O_2_]^−^, 125.0228[^1,4^A]^−^	0.30	Procyanidin B1	[41,42,43]
28	13.54	C_45_H_38_O_18_	865.1952[M−H]^−^	739.1671[M−H−C_6_H_5_O_2_−H_2_O]^−^, 713.1887[M−C_8_H_8_0_3_]^−^, 577.1301[M−H−TOP]^−^, 407.0782[M−H-TOP−C_8_H_8_O_3_−H_2_O]^−^, 289.0712[M−H−2TOP]^−^, 245.0800[M−H−2TOP−CO_2_]^−^, 125.0236[^1,4^A]^−^	3.80	Procyanidin C1	[41,42,43]
29	13.73	C_21_H_32_O_10_	443.1904[M−H]^−^	425.1931[M−H−H_2_O]^−^, 281.1394[M−H−Glc]^−^, 263.1289[M−H−Glc−H_2_O]^−^	1.13	Cynaroside A	[44]
30 *	13.96	C_16_H_14_O_5_	285.0620[M−H]^−^	151.0184[^1,3^A]^−^, 107.0291[^1,3^A-CO_2_]^−^	3.16	Isosakuranetin	[45]
31	14.15	C_15_H_20_O_10_	359.0967[M−H]^−^	197.0515[M−H−Glc]^−^, 153.0028[M−H−Glc−CO_2_]^−^, 127.0245[M−H−Glc-CO_2_−C_2_H_2_]^−^, 121.0081[M−H−Glc−CO_2_−OCH_3_]^−^	1.90	Glucosyringic acid	[46]
32	14.32	C_7_H_6_O_3_	137.0224[M−H]^−^	93.0334[M−H−CO_2_]^−^	0.30	4-Hydroxybenzoic acid	[47]
33	14.34	C_13_H_16_O_8_	299.0776[M−H]^−^	137.0241[M−H−Glc]^−^, 93.0351[M−H−Glc−CO_2_]^−^	1.2	Hydroxybenzoic acid *β*-*D*-glucose ester	[48]
34 *	14.74	C_15_H_14_O_6_	289.0724[M−H]^−^	245.0235[M−H−CO_2_]^−^, 179.0341[M−H−B ring]^−^, 167.0339[^1,2^A]^−^, 163.0385[M−H-H_2_O−B ring]^−^, 149.0234[^1,3^B]^−^, 137.0237[^1,3^A]^−^, 125.0235[^1,4^A]^−^, 109.0289[B ring]^−^	0.82	(+)-catechin	[34,36]
35	14.75	C_15_H_14_O_6_	289.0722[M−H]^−^	245.0300[M−H−CO_2_]^−^, 179.0339[M−H−B ring]^−^, 167.0340[^1,2^A]^−^, 163.0379[M−H−H_2_O−B ring]^−^, 149.0246[^1,3^B]^−^, 137.0237[^1,3^A]^−^, 125.0229[^1,4^A]^−^, 109.0199[B ring]^−^	0.56	Epicatechin	[36]
36	14.76	C_9_H_8_O_4_	179.0389[M−H]^−^	135.0472[M−H−CO_2_]^−^, 109.0440[M−H−CO_2_−CO]^−^, 89.0413[M−H−CO_2_−CO−H_2_O]^−^	3.46	Caffeic acid	[49]
37	14.91	C_15_H_20_O_9_	343.1029[M−H]^−^	181.0490[M−H−Glc]^−^, 135.0427[M−H−Glc−CH_4_O_2_]^-^^−^, 121.0286[M−H−Glc−COOCH]^−^	2.04	Methyl4-(*β*-*D*-glucopyranosyloxy)- 3-methoxybenzoate	[48]
38 *	15.27	C_16_H_18_O_9_	353.0875[M−H]^−^	191.0554[M−H−caffeoyl]^−^, 179.0365[M−H−C_7_H_10_O_5_]^−^	0.90	Chlorogenic acid	[23]
39	15.43	C_15_H_14_O_7_	305.0667[M−H]^−^	179.03325[M−H−B ring]^−^, 125.0245[^1,4^A]^−^	0.10	Epigallocatechin	[48]
40	16.13	C_12_H_22_O_8_	293.1245[M−H]^−^	131.0710[M−H−Glc]^−^	0.70	Ethyl3-(*β*-*D*-glucopyranosyloxy) butanoate	[26]
41	16.23	C_15_H_12_O_7_	303.0540[M−H]^−^	151.0051[^1,3^A]^−^, 152.0502[^1,3^B]^−^, 175.0386[M−H−H_2_O-B ring]^−^	2.50	Taxifolin	[50,51]
42	16.28	C_21_H_20_O_10_	431.1181[M−H]^−^	269.0453[M−H−Glc]^−^, 225.0671[M−H−Glc−CO]^−^, 151.0033[^1,3^A]^-^^−^, 107.01[^1,3^A−CO_2_]^−^	1.47	Cosmosiin	[52]
43	17.59	C_17_H_32_O_10_	395.1919[M−H]^−^	263.0437[M−H−Xyl]^−^, 101.0242[M−H−Xyl−Glc]^−^	0.71	Hexyl 6-*O*-*β*-*D*-Xylopyranosyl- *β*-*D*-glucopyranoside	[53]
44	17.89	C_16_H_22_O_10_	373.1128[M−H]^−^	193.0511[M−H−Glc]^−^, 149.0617[M−H−Glc−CO_2_]^−^, 123.0743[M−H−Glc−CO_2_−C_2_H_2_]^−^, 97.0547[M−H−Glc−CO_2_−2C_2_H_2_]^−^	0.27	Swertiamarin	[40]
45	18.16	C_37_H_30_O_16_	729.1441[M−H]^−^	577.1324[M−H−G]^−^, 441.0743[M−H−TOP]^−^	1.91	Procyanidin B2 3′-*O*-gallate	[41,42,43]
46	18.71	C_15_H_20_O_8_	327.1085[M−H^−^	165.0551[M−H−Glc]^−^, 147.0446[M−H−Glc−H_2_O]^−^, 119.0498[M−H−Glc−H_2_O−COOH]^−^	0.92	Diyhdromelilotoside	[54]
47	18.97	C_15_H_20_O_8_	327.1088[M−H]^−^	165.0554[M−H−Glc]^−^, 147.0447[M−H−Glc−H_2_O]^−^	0.9	Androsin	[55]
48	20.57	C_14_H_18_O_9_	329.0877[M−H]^−^	167.0338[M−H−Glc]^−^, 123.0444[M−H−Glc−CO_2_]^−^	2.33	Phloracetophenone 4’-*O*- glucoside	[28]
49	20.96	C_27_H_30_O_14_	577.1329[M−H]^−^	431.0985[M−H−Rha]^−^, 285.0244[M−H−2Rha]^−^, 256.0179[M−H−2Rha−CO]^−^	4.33	Kaempferitrin	[56]
50	23.64	C_15_H_10_O_7_	301.1189[M−H]^−^	283.1076[M−H−H_2_O]^-^^−^ 271.1089[M−H−CO_2_]^−^, 161.0470[M−H−B ring−H_2_O]^−^	0.00	Tricetin	[28]
51	24.42	C_22_H_22_O_12_	477.1014[M−H]^−^	315.0562[M−H−Glc]^−^, 300.0131[M−H−Glc−CH_3_]^−^, 151.0026[^1,3^A]^−^	1.78	Brassicin	[55]
52	24.81	C_5_H_10_O_5_	269.0647[M−H]^−^	241.0466[M−H−CO]^−^, 226.0396[M−H−C_2_H_2_O]^−^, 197.0431[M−H−CO−CO_2_]^−^	0.706	Galangin	[57]
53	24.84	C_5_H_10_O_5_	269.0650[M−H]^−^	241.0507[M−H−CO]^−^, 225.0538[M−H−CO_2_]^−^, 197.0582[M−H−CO_2_−CO]^−^, 182.0574[M−H−CO−CO_2_−CH_3_]^−^	1.67	Emodin	[58]
54	27.32	C_22_H_18_O_10_	441.0817[M−H]^−^	289.0713[M−H−G]^−^, 271.0630[M−H−G−H_2_O]^−^, 179.0362[M−H−G−B ring]^−^, 135.0242[^1,3^A]^−^, 125.0247,109.0292[B ring]^−^	2.30	(-)-Epicatechin gallate	[59]
55	27.57	C_22_H_24_O_12_	479.1198[M−H]^−^	315.0564[M−H−Glc]^−^, 211.0523[^0,4^B]^−^, 165.0558[^1,3^B]^−^, 151.0542[^1,3^A]^−^, 127.0244[B ring]^−^	0.60	3 ’- *O*-methyl-dihydroquercetin- 7-*O*-*β*-*D*-glucoside	[60]
56 *	28.18	C_28_H_24_O_16_	615.0991[M−H]^−^	463.0868[M−H−G]^−^, 301.3056[M−H−G−Glc/Gal]^−^	0.09	Quercetin-3-*O*-(6’’-galloyl)-*β*-galactopyranside/Quercetin-3-*O*-(6’’-galloyl)-*β*-glucopyranside	[61]
57	29.73	C_21_H_20_O_12_	463.0866[M−H]^−^	301.0347[M−H−Rha]^−^, 151.0022[^1,2^A−CO]^−^, 107.0149[^1,2^A−CO−CO_2_]^−^	3.40	Myricetrin	[62]
58	29.94	C_18_H_24_O_10_	399.1298[M−H]^−^	329.0508[M−H−C_5_H_10_]^−^, 169.0150[M−H−C_5_H_10_−Glc]^−^, 151.0072[M−H−C_5_H_10_-Glc−H_2_O]^−^, 125.0236[M−H−C_5_H_10_−Glc−CO_2_]^−^, 107.0142[M−H−C_5_H_10_−Glc−CO_2_−H_2_O]^−^	0.50	Taxilluside A	[12]
59 *	31.17	C_27_H_30_O_16_	609.1453[M−H]^−^	301.0354[M−H−RG]^−^, 151.0028[M−H−RG−^1,3^B]^−^	1.32	Rutin	[63]
60 *	31.30	C_21_H_20_O_12_	463.0864[M−H]^−^	301.0341[M−H−Gal]^−^, 271.0233[M−H−Gal−CHO]^−^, 151.0025[^1,3^A]^−^	3.40	Hyperoside	[62]
61 *	32.06	C_21_H_18_O_13_	477.0655[M−H]^−^	301.6332[M−H−Glc UA]^−^, 283.0230[M−H−Glc UA-H_2_O]^−^, 151.0027[^1,3^A]^−^, 107.0140[^0,4^A]^−^	4.12	Quercetin 3-*O*-*β*-*D*-glucuronide	[61]
62 *	32.46	C_21_H_20_O_12_	463.0872[M−H]^−^	301.0346[M−H−Glc]^−^ 151.0034[^1,3^A]^−^	2.2	Isoquercitrin	[64]
63 *	34.39	C_20_H_18_O_11_	433.0772[M−H]^−^	301.0354[M−H−Ara]^−^, 151.0037[^1,3^A]^−^, 107.0139[^0,4^A]^−^	0.12	Avicularin	[65]
64	34.42	C_21_H_22_O_12_	465.1035[M−H]^−^	313.0140[^1,3^B]^−^, 303.0570[M−H−Glc]^−^, 151.0391[^1,3^A]^−^, 123.0085[^1,4^A]^−^	0.774	Taxifolin 3′-*O*-*β*-*D*-glucopyranoside	[54]
65	34.99	C_27_H_30_O_15_	593.1501[M−H]^−^	285.0405[M−H−RG]^−^	1.80	Kaempferol 3-rutinoside	[58]
66	35.11	C_21_H_20_O_11_	447.0935[M−H]^−^	285.0393[M−H−Glc]^−^, 243.0497[M−H−Glc−C_2_H_2_O]^−^, 241.0341[M−H−Glc−CO_2_]^−^, 151.0029[^1,3^A]^−^	0.50	Luteolin-7-*O*-glucoside	[13]
67 *	35.38	C_21_H_20_O_11_	447.0921[M−H]^−^	301.0354[M−H−Rha]^−^, 283.0223[M−H−Rha−H_2_O]^−^, 151.0024[^1,3^A]^−^	2.7	Quercitrin	[55]
68	36.54	C_23_H_24_O_10_	459.1288[M−H]^−^	297.0762[M−H-Glc]^−^, 191.0342[M−H−Glc−B ring]^−^	1.74	8-methylretusin-7*β*-glucoside	[51]
69	36.71	C_21_H_20_O_11_	447.0925[M−H]^−^	285.0389[M−H−Glc]^−^, 151.0022[^1,3^A]^−^	1.50	Astragalin	[65]
70	37.07	C_20_H_20_O_8_	387.1141[M−H]^−^	341.1081[M−H−CH_3_-OCH_3_]^−^, 218.8840[M−H−OCH_3_−2OCH_3_-B ring]^−^, 119.0350[M−H−C_12_H_11_O_6_−CH_3_]^−^	1.30	5-Demethylnobiletin	[66]
71	37.09	C_21_H_22_O_10_	433.1131[M−H]^−^	311.0641[^1,3^A]^−^, 271.0641[M−H-Glc]^−^, 151.0031[^1,3^A]^−^, 119.0511[^1,3^B]^−^, 107.0135[B ring]^−^	0.02	Prunin	[28,55]
72	37.40	C_21_H_20_O_10_	431.0981[M−H]^−^	285.0406[M−H-Rha]^−^, 227.0352[M−H−Rha−2CO]^−^	0.60	Afzelin	[60,61]
73 *	39.51	C_15_H_10_O_7_	301.0354[M−H]^−^	273.0376[M−H-CO]^−^, 178.9988[^1,2^A]^−^, 151.0030[^1,3^A]^−^, 121.0296[^1,2^B]^−^, 107.0143[^0,4^A]^−^	0.73	Quercetin	[61]

Note: (1) *: comparison with reference standards; (2) Xyl: D-xylose; G: Gallic acid; Rib: D-ribose; Glc: D-glucose; Gal: D-galactose; Rha: L-rhamnose; Ara: L-arabinose; Glc UA: Glucuronic acid; RG: Rutinose.

**Table 2 molecules-26-06373-t002:** Identification of the differential chemical constituents.

No.	t_R_ /min	Compound	Molecular Formular	MS^1^(*m/z*)	MS^2^(*m/z*)	References
1	5.97	Monotropein	C_16_H_22_O_11_	389.1082[M−H]^−^	227.0543[M−H−Glc]^−^, 165.0544[M−H−Glc−COOH]^−^	[67]
2 *	12.57	Procyanidin B2	C_30_H_26_O_12_	577.1321[M−H]^−^	451.0989[M−H−H_2_O−C_6_H_5_O_2_]^−^, 425.0835[M−H−C_8_H_8_O_3_]^−^, 407.0733[M−H−C_8_H_8_O_3_−H_2_O]^−^, 289.0685[M−H−TOP]^−^, 245.0768[M−H−TOP−CO_2_]^−^, 179.0733[M−H−TOP−C_6_H_5_O_2_]^−^, 125.0231[^1,4^A]^−^	[41,42,43]
3	12.85	Neohesperidin dihydrochalcone	C_28_H_36_O_15_	611.1235[M−H]^−^	300.9969[M−H−Neo]^−^, 275.0183	[29]
4 *	13.22	Procyanidin B1	C_30_H_26_O_12_	577.1354[M−H]^−^	451.0986[M−H−H_2_O−C_6_H_5_O_2_]^−^, 425.0829[M−H−C_8_H_8_O_3_]^−^, 407.0745[M−H-C_8_H_8_O_3_−H_2_O]^−^, 289.0698[M−H−TOP]^−^, 245.0743[M−H−TOP−CO_2_]^−^, 179.0721[M−H−TOP−C_6_H_5_O_2_]^−^, 125.0228[^1,4^A]^−^	[41,42,43]
5 *	13.96	Procyanidin C1	C_45_H_38_O_18_	865.1952[M−H]^−^	739.1671[M−H−B ring−H_2_O]^−^, 713.1887[M−H−1,3B]^−^, 695.1411[M−H−0,3B]^−^, 577.1301,407.0782, 289.0712, 245.0800, 125.0236[1,4A]^−^	[41,42,43]
6 *	14.77	(+)-Catechin	C_15_H_14_O_6_	289.0724[M−H]^−^	245.0235[M−H−CO2]^−^, 179.0341[M−H−B ring]^−^, 167.0339[1,2A]^−^, 163.0385[M−H−H_2_O−B ring]^−^, 149.0234[1,3B]^−^, 137.0237[1,3A]^−^, 125.0235[1,4A]^−^, 109.0289[B ring]^−^	[34,36]
7	18.23	Procyanidin B2 3′-*O*-gallate	C_37_J_30_O_16_	729.1432[M−H]^−^	577.1324[M−H−Gallate]^−^, 441.0743[M−H−top-unit]^−^	[41,42,43]
8	20.93	kaempferitrin	C_27_H_30_O_14_	577.1329[M−H]^−^	430.0985[M−H−Rha]^−^, 283.0244[M−H−2Rha]^−^, 256.0179[M−H−2Rha−CO_2_]^−^	[56]
9 *	27.26	(−)-Epicatechin gallate	C_22_H_18_O_10_	441.0817[M−H]^−^	289.0713[M−H−C_7_H4O4]^−^, 271.0630[M−H−C_7_H4O4−H_2_O]^−^, 179.0362[M−H−C_7_H4O4−B ring]^−^, 135.0242[1,3A]^−^, 125.0247, 109.0292[B ring]^−^	[59]
10 *	31.39	Hyperoside	C_21_H_20_O_12_	463.0859[M−H]^−^	301.0341[M−H−Glc]^−^, 271.0233,151.0025[1,3A]^−^	[62]
11	31.95	Astilbin	C_21_H_22_O_11_	449.1077[M−H]^−^	431.0945[M−H−H_2_O]^−^, 303.0478[M−H−Rha]^−^, 285.0380[M−H−Rha−H_2_O]^-^, 178.9977[M−H−Rha−H_2_O−B ring ]^−^, 151.0030, 125.0240	[68]
12 *	32.24	Quercetin 3-*O*-*β*-*D*-glucuronide	C_21_H_18_O_13_	477.066[M−H]^−^	301.6332[M−H−Glc UA]^−^, 283.0230[M−H−GlcA−H_2_O]^−^, 151.0027[1,3A]^−^, 107.0140[B ring]^−^	[61]
13	32.39	Acteoside	C_29_H_36_O_15_	623.1976[M−H]^−^	461.1651[M−H−C_9_H_6_O3]^−^, 315.1087[M−H−C_9_H_6_O3−C_6_H_10_O4]^−^, 179.0340, 161.0232, 135.0444	[69]
14 *	32.88	Isoquercitrin	C_21_H_20_O_12_	463.0866[M−H]^−^	301.0346[M−H−Glc]^−^, 151.0034[1,3A]^−^	[64]
15	34.48	Narcissin	C_28_H_32_O_16_	623.1975[M−H]^−^	461.1654[M−H−Rha]^−^, 315.1075[M−H−Rha−Glc]^−^	[70]
16	35.19	Isoastilbin	C_21_H_22_O_11_	449.1073[M−H]^−^	431.0945[M−H−H_2_O]^−^, 303.0478[M−H−Rha]^−^, 285.0380[M−H−Rha−H_2_O]^−^, 178.9977[M−H−Rha−H_2_O−B ring ]^−^,151.0030,125.0240	[68]
17 *	35.35	Quercitrin	C_21_H_20_O_11_	447.0924[M−H]^−^	301.0354[M−H−Rha]^−^, 283.0223[M−H−Rha−H_2_O]^−^, 151.0024[M−H−Rha-1,3B]^−^	[55]
18	35.54	Isohemiphloin	C_21_H_22_O_10_	433.1131[M−H]^−^	269.0442[M−H−Glc]^−^, 178.9977[M−H−Glc−B ring]^−^, 151.0028[^1,3^A]^−^	[71]
19	36.48	(−)-Epiafzelechin 3-*O*-gallate	C_22_H_18_O_9_	425.0874[M−H]^−^	273.0478[M−H−C_7_H4O4]^−^, 151.0390[^1,3^A]^−^, 137.0234, 125.0237[^1,4^A]^−^	[72]
20	36.83	Rhamnitrin	C_22_H_22_O_11_	461.0713[M−H]^−^	312.9958[M−H−Rha]^−^, 285.0016[M−H-Rha−OCH_3_]^−^	[73]
21	37.89	Cascaroside A	C_27_H_32_O_14_	579.2082[M−H]^−^	371.1496[M−H−Glc−CH_2_OH]^−^, 256.1267[M−H−2Glc]^−^, 228.0413[M−H−2Glc−CO]^−^	[74]
22	38.64	Apigenin-7-*O*-rutinoside	C_27_H_30_O_14_	577.0974[M−H]^−^	415.0855[M−H−Rha]^−^, 269.0745[M−H−Rha−Glc]^−^, 225.0675[M−H−Rha-Glc−CO]^−^	[75]
23	38.85	Apimaysin	C_27_H_28_O_13_	559.1580[M−H]^−^	397.0178[M−H−Rha]^−^, 280.4159[M−H−C_12_H_18_O8]^−^	[76]

Note: (1) *: comparison with reference standards; (2) Glc: D-glucose; Rha: L-rhamnose; Ara: Glc UA: Glucuronic acid.

**Table 3 molecules-26-06373-t003:** Information of Taxilli Herba samples from 7 different hosts.

No.	Family	Hosts	Producing Area	Harvest Time
S1	Moraceae	*Morus alba* L.	Wuzhou Guangxi	2020.12.28
S2	Hamamelidaceae	*Liquidambar formosana* Hance.	Wuzhou Guangxi	2020.12.28
S3	Aquifoliaceae	*Ilex latifolia* Thunb.	Wuzhou Guangxi	2019.5.19
S4	Rosaceae	*Crataegus pinnatifida Bge. var. major* N.E.Br.	Wuzhou Guangxi	2019.5.19
S5	Passifloraceae	*Passiflora edulia* Sims.	Wuzhou Guangxi	2019.5.20
S6	Rosaceae	*Pyrus pyrifolia* (Burm. F.) Nakai.	Wuzhou Guangxi	2019.5.19
S7	Lauraceae	*Cinnamomum camphora* (L.) Presl.	Gongcheng Guangxi	2020.12.21

## Data Availability

The data presented in this study are available in Appendix A.

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
