# Peer review of "Qualitative Analysis and Componential Differences of Chemical Constituents in Taxilli Herba from Different Hosts by UFLC-Triple TOF-MS/MS"

_molecules, 2021, doi:10.3390/molecules26216373_

Round 1
Reviewer 1 Report
The authors performed the characterization of chemical constituents in TH from 7 different hosts was analyzed by ultra-fast liquid chromatography coupled with triple quadrupole-time of flight tandem mass spectrometry (UFLC-Triple TOF-MS/MS. The authors they identified 73 and found 23 differential characteristic constituents.
Line 59-64. Remove paragraph. It is not an article on analytical methods
Line 77-80. Change this paragraph to methodology.
Line 84-86. Change this paragraph to methodology.
Review pages 3, 4 and 5, there is a table1 repeated.
In methodology, the authors do not mention the library with which the equipment was equipped.
Author Response
The authors performed the characterization of chemical constituents in TH from 7 different hosts was analyzed by ultra-fast liquid chromatography coupled with triple quadrupole-time of flight tandem mass spectrometry (UFLC-Triple TOF-MS/MS. The authors they identified 73 and found 23 differential characteristic constituents.
Reply: First of all, thank you very much for your recognition and encouragement to our work. We appreciate for your warm work earnestly.
In order to fulfill the standards be set by Journal, we revised the article on the basis of reviewers’ comments. And hope that the correction will meet with approval.
Once again, thank you very much for your suggestions.
Point 1: Line 59-64. Remove paragraph. It is not an article on analytical methods.
Reply 1: Thanks for your professional comments.
We considered not remove the line 59-64 with following reasons. LC-MS, especially UFLC-Triple TOF-MS/MS has two advantages of high separation ability of chromatography and high discrimination ability of mass spectrometry, and it is often used for identification of multiple components in traditional Chinese medicine. It was the main method used in our article and the data obtained by this method can help us to identify and speculate the chemical composition of Taxilli Herba. Therefore, we would like to make a brief mention in the introduction of this article, and it would be convenient for readers to understand.
Our paper is truly an article on application research rather than analytical methods.
Point 2: Line 77-80. Change this paragraph to methodology.
Reply 2: Thank you for your sincere advice.
Line 77-80 (Line 87-90 in revised version) of Optimization of Extraction Conditions included both discussion and result. Before submission, we refer to the submission requirements of Molecules and other better journals. It is found that this part is generally placed in the Result part, so we would like to put these two parts in the Result part.
Point 3: Line 84-86. Change this paragraph to methodology.
Reply 3: Thank you for your sincere advice.
Line 84-86 (Line 95-98 in revised version) of Optimization of UFLC-Triple TOF-MS/MS Conditions included both discussion and result. Before submission, we refer to the submission requirements of Molecules and other better journals. It is found that this part is generally placed in the Result part, so we would like to put these two parts in the Result part.
Cai, Z.C.; Wang, C.C.; Zou, L.S.; Liu, X.H.; Chen, J.L.; Tan, M.X.; Mei, Y.Q.; Wei, L.F. Comparison of Multiple Bioactive Constituents in the Flower and the Caulis of Lonicera japonica Based on UFLC-QTRAP-MS/MS Combined with Multivariate Statistical Analysis. Molecules. 2019, 24, 1936.
Chen, C.H.; Wang, C.C.; Liu, Z.X.; Liu, X.H.; Zou, L.S.; Shi, J.J.; Chen, S.Y.; hen, J.L.; Tan, M.X. Variations in Physiology and Multiple BioactiveConstituents under Salt Stress Provide Insight into the Quality Evaluation of Apocyni Veneti Folium. Molecules. 2018, 19, 3042.
Zhao, H.; Yan, Y.; Chai, C.; Zou, L.S.; Liu, X.H.; Wang, S.N.; Hua, Y.J. Dynamic Changes of Eight Bioactive Constituents in Magnoliae Officinalis Cortex Based on UFLC-QTRAP-MS/MS Combined with Grey Relational Analysis. Curr Pharm Anal. 2019, 15, 497–504.
Zhao, X.Q.; Wang, W.S.; Zhang, F.; Deng, J.L.; Fu, B.Y. Comparative metabolite profiling of two rice genotypes with contrasting salt stress tolerance at the seedling stage. PLoS ONE. 2014, 25, 748–752.
Yan, Y.; Cai, H.; Zhao, H.; Chen ,C.H.; Liu, Z.X.; Liu, X.H.; Chai, C.; Wang,.S.N.; Hua, Y.J. Quality evaluation of Eucommiae Cortex processed by different methods and “sweating” conditions based on simultaneous determination of multiple bioactive constituents combined with gray relational analysis. Sep Sci. 2018, 41, 1050–1062.
Point 4: Review pages 3, 4 and 5, there is a table1 repeated.
Reply 4: Thanks for your kind reminder and we are very sorry for our negligence. Table 1 on page 3-4 was deleted this time.
Table 1. Identification of 73 constituents in Taxilli Herba by UFLC-Triple TOF-MS/MS.
No. |
tR /min |
Molecular formula |
MS1(m/z) |
MS2(m/z) |
Error /ppm |
Compound |
References |
1 |
2.24 |
C6H12O6 |
179.0561[M-H]- |
161.0457[M-H-H2O]-, 117.0207, 89.0248, 71.0158 |
0.05 |
Glucose |
[15] |
2 |
2.27 |
C5H10N2O3 |
145.0633[M-H]- |
128.0539[M-H-NH3]-, 127.0514[M-H-H2O]-, 101.0725[M-H-CO2]-, 99.0560[M-H-HCOOH]- |
0.70 |
Glutamine |
[16] |
3 |
2.54 |
C11H20O10 |
311.0991[M-H]- |
233.0654[M-H-C2H3O2-CHOH]-,173.0446[M-H-C2H3O2-CHOH-C2H4O2]-, 131.0341[M-H-C6H11O6]-,99.0095[M-H-Xyl-C3H6O2]-,71.0155[M-H-Xyl-C4H8O3]- |
1.93 |
Primeverose |
[17] |
4* |
2.59 |
C10H13N5O6 |
282.0830[M-H]- |
150.0426[M-H-Rib]-, 133.0153[M-H-Rib-H2O]- |
4.90 |
Guanosine |
[18] |
5 |
2.63 |
C7H12O6 |
191.0057[M-H]- |
173.0454[M-H-H2O]-, 127.0397[M-H-H2O-HCOOH]- ,59.0160, 71.0161, 85.0304 |
0.05 |
Quinic acid |
[19] |
6 |
2.86 |
C7H10O5 |
173.0472[M-H]- |
155.0407[M-H2O]-, 137.0246[M-H-2H2O]-,129.0189[M-H-CO2]-,111.0451[M-H-CO2-H2O]- |
1.10 |
Shikimic acid |
[20] |
7 |
3.12 |
C7H6O2 |
133.0146[M-H]- |
115.0041[M-H-H2O]-,71.0160[M-H-H2O-CO2]- |
2.60 |
Malic acid |
[21] |
8 |
4.96 |
C6H8O7 |
191.0206[M-H]- |
173.0101[M-H-H2O]-,154.9982[M-H-2H2O]-,129.0187[M-H-H2O-CO2]-,103.0400[M-H-2CO2]- |
0.00 |
Citric acid |
[22] |
9* |
7.93 |
C7H6O5 |
169.0138[M-H]- |
125.0240[M-H-CO2]-,107.0141[M-H-CO2-H2O]-,97.0341[M-H-CO2-CO]-,69.0374[M-H-CO2-2CO]- |
2.60 |
Gallic acid |
[23,24] |
10 |
8.00 |
C19H16O4 |
307.1029[M-H]- |
145.0508,127.0397 |
2.60 |
Bisdemethoxycurcumin |
[25] |
11 |
8.86 |
C13H16O10 |
331.0673[M-H]- |
179.0137[M-H-G]-,169.0253,161.024[M-H-G-H2O]- |
1.80 |
Glucogallin |
[24] |
12 |
9.03 |
C13H18O8 |
301.0928[M-H]- |
139.0324[M-H-Glc]-,123.0089[M-H-Glc-O]- |
1.66 |
Tachioside |
[22] |
13 |
9.34 |
C11H20O9 |
295.1045[M-H]- |
131.0934[M-H-Glc]-,113.0252[M-H-Glc-H2O]-,85.0306, 71.016, 59.0162 |
0.30 |
(1S)-2(Acetyloxy)-1-(hydroxymethy)ethyl-β-D-glucopyranoside |
[26] |
14 |
9.39 |
C20H20O11 |
435.1129[M-H]- |
271.0448[M-H-Glc]-,313.0354, 151.0037, 125.0245 |
2.52 |
Homomangiferin |
[27] |
15 |
9.43 |
C24H20O8 |
435.1124[M-H]- |
313.0509,151.0025,123.0085 |
0.23 |
Isochinomin |
[28] |
16 |
9.55 |
C26H22O10 |
493.1183[M-H]- |
331.0654[M-H-Gal]- |
1.42 |
Fluorescein-β-D-galactopyranoside |
[25] |
17 |
10.16 |
C23H18O7 |
405.1031[M-H]- |
169.0134,71.0154 |
4.93 |
Toddacoumaquinone |
[28] |
18 |
10.31 |
C13H16O10 |
331.1061[M-H]- |
169.0134[M-H-Glc]-,125.0240[M-H-Glc-CO2]- |
0.60 |
Gallic acid 3-O-β-D- glucopyranoside |
[29–31] |
19 |
10.77 |
C22H16O5 |
375.0927[M-H]- |
285.0437[M-H-Benzyl group]-,151.0030[1,3A]- |
1.06 |
7-O-Benzyl Luteolin |
[23] |
20 |
11.12 |
C24H20O7 |
419.1177[M-H]- |
271.0421,151.0033 |
3.50 |
Artonol B |
[32] |
21 |
11.13 |
C25H20O9 |
463.0869[M-H]- |
435.0761[M-H-CO]-,273.0364[M-H-C10H11O3]-,151.0927[1,3A]- |
0.00 |
Hydrocarpin |
[33,34] |
22 |
11.30 |
C26H30O14 |
565.1547[M-H]- |
403.0473[M-H-Glc]-,241.0241[M-H-2Glc]- |
2.80 |
Mulberroside F |
[35] |
23* |
11.38 |
C7H6O4 |
153.0194[M-H]- |
109.0303[M-H-CO2]-, 101.0314[M-H-CO2-CO]-, |
6.90 |
Protocatechuic acid |
[23] |
24 |
11.68 |
C12H14O9 |
301.0565[M-H]- |
283.0456[M-H-H2O]-,125.0241[M-H-Glc UA]-,107.0144[M-H-Glc UA-H2O]- |
0.00 |
2,6-Dihydroxyphenyl-β-D- glucopyranosiduronic acid |
[36] |
25 |
12.24 |
C9H8O3 |
163.0401[M-H]- |
119.0482[M-H-CO2]-,93.0316[M-H-CO2-C2H2]- |
0.18 |
p-Coumaric acid |
[19] |
26 |
12.57 |
C30H26O12 |
577.1354[M-H]- |
451.0989[M-H-H2O-C6H5O2]-,425.0835[M-H-C8H8O3]-,407.0733[M-H-C8H8O3-H2O]-, 289.0685[M-H-TOP]-,245.0768[M-H-TOP-CO2]-,179.0733[M-H-TOP-C6H5O2]-,125.0231[1,4A]- |
0.40 |
Procyanidin B2 |
[37–39] |
Point 5: In methodology, the authors do not mention the library with which the equipment was equipped.
Reply 5: Thanks for your professional comments.
- We added the information “(H1650-W high speed centrifuge, Hunan Xiangyi Laboratory Instrument Development Co., Ltd., Hunan, China)”of the instrument in Line 77-78 of Page 22.
- In the “Chromatographic Processing and Statistical Analysis”part, “Based on the above qualitative results, PLS-DA was used to perform dimensionality reduction analysis on the data to obtain information about differences between groups. ” was corrected into “Based on the above qualitative results, PLS-DA using the SIMCA-P 13.0 software (Umetrics AB, Umea, Sweden) was used to perform dimensionality reduction analysis on the datato obtain information about differences between groups. ” in Line 110-111 of Page 23.
- In the “Identification of the Differential Chemical Components”part, “Except for comparison with the components in Table 1, other unknown differential components were identified through databases and literature.” was corrected into “Except for comparison with the components in Table 1, other unknown differential components were identified through literature and databases including SciFinder (https://scifinder.cas. org/), HMDB (https://hmdb.ca/), CNKI (https://kns.cnki.net/).” in Line 117-118 of Page 23.
We appreciate for reviewers’ warm work earnestly, and hope that the correction will meet with approval.
Once again, thank you very much for your comments and suggestions.
Reviewer 2 Report
Review on
Qualitative Analysis and Componential Differences of Chemical Constituents in Taxilli Herba from Different Hosts by UFLC-Triple TOF-MS/MS
In this article, an LC-MS and tandem mass spectrometric study of Taxilli Herba is published from different hosts. Numerous compounds have been identified and further analysis was done to identify differential compounds such as partial least squares discriminant analysis.
The article is well organized only some additional details should be added (see the comments). The results are promising and they are presented properly. A lot of work has been done during the analysis The article is relatively long, supplementary material would be beneficial. However, it is just my personal opinion it should not be corrected
I suggest minor revision before acceptance.
At the begging of the article, the identification of compounds is written. Many compounds are identified however, it is not indicated which host result in these compounds. In Figure 1. 73 peaks are indicated. Was it the maximum number of compounds detected/identified in the Taxilli Herba from the different hosts? In the text, it should be added how many identified compounds are in the different samples.
It should be added which compound correspond to which host as an additional row in table 1.
Fig 2 should be separated, since the first part contains exact structures, while the second part contains tables and general structures.
Comment:
- The beginning of Table 1. is repeated up to 26. Compound. Delete the unnecessary part
- Page 11 line 2. DRA is written instead of RDA, please correct.
- Figure 5. Please increase the letter size, since it is really hard to read.
Author Response
Review on
Qualitative Analysis and Componential Differences of Chemical Constituents in Taxilli Herba from Different Hosts by UFLC-Triple TOF-MS/MS
In this article, an LC-MS and tandem mass spectrometric study of Taxilli Herba is published from different hosts. Numerous compounds have been identified and further analysis was done to identify differential compounds such as partial least squares discriminant analysis.
The article is well organized only some additional details should be added (see the comments). The results are promising and they are presented properly. A lot of work has been done during the analysis The article is relatively long, supplementary material would be beneficial. However, it is just my personal opinion it should not be corrected.
I suggest minor revision before acceptance.
Point 1: At the begging of the article, the identification of compounds is written. Many compounds are identified however, it is not indicated which host result in these compounds. In Figure 1. 73 peaks are indicated. Was it the maximum number of compounds detected/identified in the Taxilli Herba from the different hosts? In the text, it should be added how many identified compounds are in the different samples.
Reply 1:Thanks for your professional advice. We are so sorry to make it unclear.
- We selected the samples where the most common host is Morus alba for analysis. These compounds identified or speculated were all in Taxilli Herba from Morus albaL.. In the supplementary material, we have added table S1 of the identification of the chemical constituents in samples from different hosts for better understanding.
- In Figure 1, there were more than 73 peaks on the mass spectrometry. Some of the peaks corresponding to the compounds had not been identified due to the following reasons: the current plant metabolite database is not very complete, and the 73 compounds were the maximum number of compounds that we could currently identify based on the compound cleavage method and standards. Therefore, 73 compounds were the maximum number we could identified by UFLC-Triple TOF-MS/MS, which were all contained in Taxilli Herba from Morus alba.
- 85 constituents were identified in the different samples from 7 hosts, and the specific results were shown in Table S1in the supplementary materials, and the information was added in Line 186-187 of Page 16.
Point 2: It should be added which compound correspond to which host as an additional row in table 1.
Reply 2: Thank you for your sincere advice. We thought it is more appropriate to revised the title of Table 1 “Identification of 73 constituents in Taxilli Herba by UFLC-Triple TOF-MS/MS.” to “Identification of 73 constituents in Taxilli Herba from Morus alba L. by UFLC-Triple TOF-MS/MS.” on Page 3. At the same time, we added the specific information shown in Table S1 in the supplementary materials, and which compound corresponding to which host was clearly presented in Table S1.
Point 3: Fig 2 should be separated, since the first part contains exact structures, while the second part contains tables and general structures.
Reply 3: Thank you for your kind reminder.We divided Figure 2 on Page 8-9 into 2 parts to make it clearer and easier to understand, where part A represented the exact structures, and part B represented the table and general structures in Line 5 of Page 9.
Point 4: The beginning of Table 1. is repeated up to 26. Compound. Delete the unnecessary part.
Reply 4:Thanks for your kind advice and we are very sorry for our negligence. The unnecessary part in Table 1 on page 3-4 was deleted.
Point 5: Page 11 line 2. DRA is written instead of RDA, please correct.
Reply 5: Thanks for your kind reminder and we are very sorry for our negligence. Page 10 line 28 ( (Line 29 of Page 9 in revised version)) “DRA”was corrected into “RDA”.
Point 6: Figure 5. Please increase the letter size, since it is really hard to read.
Reply 6: Thanks for your professional comments. We have adjusted the letter size of the Figure 5 to make it convenient to read.
Once again, thank you very much for your comments and suggestions.
Reviewer 3 Report
My comments, are in the attached document, please check

Author Response
Point 1: In the introduction, over the last paragraph, talk about methodology and results, please check out this, i will hope rather,to talk more about PLS-DA, some background where it has been applied.
Reply 1: Thanks for your professional advice. We have added some examples of the application of PLS-DA in Chinese medicine in the last paragraph of the introduction, and the specific content has been highlighted in the text in Line 68-74 of Page 2.
“PLS-DA is a supervised statistical method of discriminant analysis, which can be used to establish a model of the relationship between the expression of metabolites and the sample category to realize the prediction of the sample category. At present, PLS-DA is widely used in the quality control of traditional Chinese medicines, such as the authenticity identification of medicinal materials, the identification of base sources, and the rapid identification of medicinal materials of different origins[15–18].”
[15]Wong, Ka.H.; Valentina Razmovski-Naumovski.; Li, K.M.; George Q, Li.; Kelvin, Chan. Differentiating Puerariae Lobatae Radix and Puerariae Thomsonii Radix using HPTLC coupled with multivariate classification analyses. J Pharm Biomed Anal. 2014, 95, 11-19.
[16]Sun, L.L.; Ren, X.L.; Zhang, H.J.; Wang, M.; Liu, Y.N.; Deng, Y.R.; Qi, A.D. Research on processing of Radix Polygoni Multiflori based on UPLC fingerprints and chemometrics. Chin J Tradit Chin Med Pharm. 2017, 32, 2194-2197.
[17]Zhou, X.D.; Tang, L.Y.; Wu, H.W.; Zhou, G.H.; Wang, T.; Kou, Z.Z.; Li, S.X.; Wang, Z.J. Chemometric analyses for the characterization of raw and processed seeds of Descurainia sophia (L.) based on HPLC fingerprints. J Pharm Biomed Anal. 2015, 111, 1-6.
[18]Zhang, D.K.; Han, X.; Li, R.Y.; Niu, M.; Zhao, Y.L.; Wang, K.B.; Yang, M.; Xiao, X.H. Analysis on characteristic constituents of crude Aconitum carmichaelii in different regions based on UPLC-Q-TOF-MS. China J. Chin. Mater. Med. 2016, 41, 463-469.
Point 2: Check out the table 1, there is repeated information.
Reply 2:Thanks for your kind advice and we are very sorry for our negligence. The unnecessary part In Table 1 on page 3-4 was deleted.
Point 3: Because you used the ion source in negative mode, only, however you report compounds which in la ionization are positive, can you explain. The other things is that you should verify the identification of the compounds with some references material and is not clear that you do that.
Reply 3: Thanks for your kind advice.
- We used the ion source in negative mode rather than in positive mode for the following reasons: on the one hand, flavonoids are the main components contained in Taxilli Herbaaccording to previous literature. On the other hand, flavonoids are acidic due to the phenolic hydroxyl groups in their structures, and more prone to lose protons to form negative ions. At the same time, flavonoids are also more sensitive in the negative ion mode, and the negative ion mode in the mass spectrometry is more suitable for the analysis of acidic samples as well.
- Among the compounds we identified, 15 components were verified with reference standards marked with “*”in table 1, and the rest were inferred based on literature and databases.
Point 4: In my opinion is quite large la conclusion, check out this.
Reply 4: Thanks for your professional advice.We have tried our best to make some revision to the conclusion as follows: “ In our study, an efficient method based on UFLC-Triple TOF-MS/MS was established for the qualitative characterization of Taxili Herba from Morus alba L. The results showed that 73 constituents were identified in total, including flavonoids and phenolic acids, etc. The fragmentation pathways of flavonoids, phenolic ccids, phenylpropanoids, tannins and glycosides were preliminarily deduced by the fragmentation behavior of these constituents. Simultaneously, the results of PLS-DA showed that TH samples from Morus alba L and other hosts were clearly separated. 23 differential characteristic constituents were screened based on PLS-DA scores plot and VIP plot, and three common differential components showed different changing laws. In a word, the results could helps us have a clearer understanding of the chemical constituents of TH and reveal differential components in TH from different hosts. The findings will contribute to comprehensive evaluation and intrinsic quality control of TH and provide a scientific basis for the identification of TH from different hosts. ”
Once again, thank you very much for your comments and suggestions.
Reviewer 4 Report
This research has a lot of work, but the presentation should be improved a lot. Working with polyphenols is challenging because of the amount of data, and therefore information, to manage. Although the work is interesting it is quite difficult for the reader to understand what has been accomplished: there is a lot of information but it is not well organised. Tables and figures do not give good information; they should be presented as supplementary information.
On the other hand, the statistical treatment of data is not well described: it should be more extensively described.
Discussion section is not a real discussion, but a gathering of different information collected through performed analysis.
Author Response
Point 1: This research has a lot of work, but the presentation should be improved a lot. Working with polyphenols is challenging because of the amount of data, and therefore information, to manage. Although the work is interesting it is quite difficult for the reader to understand what has been accomplished: there is a lot of information but it is not well organised. Tables and figures do not give good information; they should be presented as supplementary information.
Response 1: Thanks for your professional advice. We have tried our best to revise the manuscript to make it easier for the readers to understand what has been accomplished.
- We mainly carried out two aspects of research and analysis.
The first part was the identification and speculation of the chemical constituents of Taxilli Herba using UFLC-Triple TOF-MS/MS. Morus alba L. was currently the most common host of Taxilli Herba on the market. We chose to analyze the chemical constituents of Taxilli Herba from Morus alba L., which helps us to have a deeper understanding of this medicine material. The findings would contribute to comprehensive evaluation and intrinsic quality control of Taxilli Herba as well. In the process of data processing, we identified and speculated on the compounds as much as possible by comparing with standards and searching the related databases and literature, and finally identified 73 compounds. Table 1 in the article presented the results of constituents identification, including the retention time, molecular formula, MS spectrometry, and MS/MS spectrometry fragmentation information. Figure 2 included the exact structure and general structure of compounds involved in the summary of the cleavage law. At the same time, the mass spectroscopy cleavage behavior of 5 types of compounds were summarized including flavonoids, phenolic acids, phenylpropanoids, tannins and glycosides. Figure 4A-G showed the possible cleavage pathways of the more typical compounds in the above five types of compounds, which would help us to have an overall understanding of the mass spectrometry fragmentation behaviour of this type of compounds.
The second part was that we chose Taxilli Herba samples from 7 hosts for the analysis of the differential components, including Morus alba L. (a), Liquidambar formosana Hance. (b), Ilex latifolia Thunb. (c), Crataegus pinnatifida Bge. var. major N.E.Br. (d), Passiflora edulia Sims. (e), Pyrus pyrifolia (Burm. F.) Nakai. (f), and Cinnamomum camphora (L.) Presl (g). PLS-DA was applied to distinguish the samples from different hosts and reveal the differential characteristic constituents based on the importance in projection (VIP) and p-value. Differential characteristic constituents were identified by comparing with the components in table 1 and searching the related databases and literature. The identification of the differential chemical constituents was shown in Table 2, and PLS-DA scores plot and VIP score plot of TH samples from different hosts were shown in Figure 5. Figures and tables could help us understand the content of the article more intuitively and clearly.
(2)Figures and tables in the text were the specific presentation of our experimental results. Table 1 in the article presented the results of constituents identification , including the retention time, molecular formula, MS spectrometry, and MS/MS spectrometry fragmentation information. Figure 2 included the exact structure and general structure of some compounds involved in the summary of the cleavage law. Figure 4A-G showed the possible cleavage pathways of the more typical compounds in the above five types of compounds, which would help us to have an overall understanding of the mass spectrometry fragmentation behaviour of this type of compounds. The identification of the differential chemical constituents in samlpes from 7 hosts was shown in Table 2, and PLS-DA scores plot and VIP score plot of TH samples from different hosts were shown in Figure 5. These figures and tables would more intuitively help readers understand the work and results we have done. These tables and figures were the main content of the article, so we preferred to put them in the main text rather than in the supplementary materials.
Point 2: On the other hand, the statistical treatment of data is not well described: it should be more extensively described.
Response 2: Thank you for your sincere advice. We have tried our best to organize and describe the data to make it clearer for readers. We hope that the corrections would meet with approval.
- In the first part of dentification and speculation of the chemical constituents of Taxilli Herba, in the process of data processing, first of all, a database on chemical constituents of Taxilli Herba was established by searching for relevant documents at home and abroad, and the data were put into the Peakview 1.2 (Sciex AB, Framinghan, MA, USA). Secondly, the target compounds were screened according to the XIC Manager function in the software, and themolecular formula of the compound corresponding to each chromatographic peak based on the principle that the actual and theoretical relative molecular mass error is less than 10×10−6 and combined with the isotope abundance ratio were determined. Thirdly, the inferred results were confirmed by comparing the retention time (tR) and mass spectra of reference substances. Fourthly, the remaining compounds without standards were speculated by the database established in the previous period and the related literature and database including SciFinder (https://scifinder.cas. org/), HMDB (https://hmdb.ca/), CNKI (https://kns.cnki.net/). In the end, ChemDraw Ultra 7.0 combined with Snagit 11.0 software were used to draw the derivation diagram of the compound cleavage pathway.
In the second part of analysis of differential components in TH from different hosts, the mass spectrometry data processed by Peakview 1.2 (Sciex AB, Framinghan, MA, USA) and Markerview 1.2.1 (Sciex AB, Framinghan, MA, USA) software were imported into SIMCA-P 13.0 (Umetrics AB, Umea, Sweden) software for analysis. Based on the qualitative results, PLS-DA using the SIMCA-P 13.0 software (Umetrics AB, Umea, Sweden) was used to perform dimensionality reduction analysis on the datas to obtain information about differences between groups. The difference chemical components of TH from different hosts were found according to the VIP and p-value obtained by the PLS-DA model. Except for comparison with the components in Table 1, other unknown differential components were identified through literature and databases including SciFinder (https://scifinder.cas. org/), HMDB (https://hmdb.ca/), CNKI (https://kns.cnki.net/).
- Table 1 in the article presented the results of constituents identification , including the retention time, molecular formula, MS spectrometry, and MS/MS spectrometry fragmentation information. Figure 1 was the base peak chromatogram (BPC) of Taxilli Herba from Morus alba. in negative ion mode. Figure 2 included the exact structure and general structure of some compounds involved in the summary of the cleavage law. Figure 3 contained Several RDA cleavage modes of flavonoids in negative ion mode. Figure 4A-G showed the possible cleavage pathways of the more typical compounds in the above five types of compounds, which would help us to have an overall understanding of the mass spectrometry fragmentation behaviour of this type of compounds. The identification of the differential chemical constituents of samlpes from 7 hosts was shown in Table 2, and PLS-DA scores plot and VIP score plot of TH samples from different hosts were shown in Figure 5.
The data after processed was shown in tables and figures, which helps us to have an overall understanding of the chemical composition of Taxilli Herba from Morus alba L., and the significantly different constituents that were screened out could be used as chemical markers to distinguish Taxilli Herba from Morus alba L. and other hosts.
Point 3: Discussion section is not a real discussion, but a gathering of different information collected through performed analysis.
Response 3: Thanks for your professional comments.We have tried our best to make some revision to the discussion as follows: “As mentioned previously, Taxilli Herba is a semi parasitic plant with complex hosts. The demand for TH in clinical is gradually increasing as well. In recent years, there have been few research reports on the chemical composition of TH. What’s more, the TH from different hosts currently circulating on the market are difficult to distinguish based on their appearance. In our study, we tried to establish a methodology to exploring the chemical constituents in TH. There were 73 chemical constituents identified finally in TH from Morus alba L., and flavonoids were the main component (Table 1). The scores scatter plot of PLS-DA showed that the samples from Morus alba L. and other hosts were significantly divided into two groups (Figure 5). 23 differential chemical constituents were initially identified of samples from 7 hosts, and the relative contents of three common differential constituents of quercetin 3-O-β-D-glucuronide, quercitrin and hyperoside in TH from Morus alba L. were higher than that of samples from other hosts (Figure 6). The results revealing possible components in TH will help us to have a deeper understanding of this medicine material, and can also be used as a basis for distinguishing samples of TH from different hosts. As far as the current situation is concerned, the diversified sources of medicinal materials are an important reason for the uneven quality of TH. At present, there are many medicinal materials from different host plants on the market, and TH from Morus alba L. is the most widely used clinically. However, the impact of the hosts on the quality of the medicinal materials in many aspects is still unknown. Systematic research on multiple levels from ingredients to curative effects to explain whether the effects of TH from different hosts are the same or different is also a question worthy of discussion. The most important thing is that this study could provide basic information for the quality formation of TH.”
Once again, thank you very much for your comments and suggestions.
Round 2
Reviewer 4 Report
Although there have been improvements, they are not enough. English style must be improved a lot more. In some paragraphs it is not easy to understand the meaning. In my opinion, the authors should improve the article completely.
Some advices:
*When an abbreviation is used in the abstract, the same process must be followed in the text: the whole words and then the abbreviation.
*L. 87: "solid-liquid ratio", but then it says v/v
*Do not say "it's" or expressions like that, but "it is"
*L. 93 and so on: what is the meaning of "formic acid water"?L. 100: what is "sample (S1-4)"?
*What do the authors mean exactly when they say "speculate"?
*Section 2.3.1, L. 11: Fig. 3 should go in this line
*In section 2.3.1, I do not see all the sub-sections for the 33 flavonoids, namely: dihydroflavones, dihydrofalvonols, falvonols, isoflavones, flavones, flavones and other flavonoids.
*What do the authors mean exactly when they say "differential" throughout the manuscript?
Author Response
Although there have been improvements, they are not enough. English style must be improved a lot more. In some paragraphs it is not easy to understand the meaning. In my opinion, the authors should improve the article completely.
Reply : First of all, thank you very much for your recognition and encouragement to our work. We appreciate for your warm work earnestly. We have tried our best to improve the English style of the manuscript to make it easier to understand. We hope that the correction would meet with approval.
Point 1: When an abbreviation is used in the abstract, the same process must be followed in the text: the whole words and then the abbreviation.
Reply 1: Thanks for your professional comments. We have checked the full text and made the following changes:
- We have added the whole word “traditional Chinese medicine”before “TCM” in Line 63 of Page 2.
- We have added the whole word “variable importance in projection”before “VIP” in Line 80-81 of Page 2.
Point 2: L. 87: "solid-liquid ratio", but then it says v/v
Reply 2: Thanks for your kind advice and we are very sorry for our negligence.
The unit of solid-liquid ratio were corrected to “w/v” in revised part of Line 90 of Page 2.
Point 3: Do not say "it's" or expressions like that, but "it is"
Reply 3: Thanks for your professional comments.
Page 2 Line 56 “it’s” was corrected into “it is”.
Point 4: L. 93 and so on: what is the meaning of "formic acid water"?L. 100: what is "sample (S1-4)"?
Reply 4: Thank you for your sincere advice.
- “formic acid water”means “4% (v/v) formic acid water solution”, and we have corrected “formic acid water” into “0.4% (v/v) formic acid water solution” in Line 96-101 of Page 3 in revised part.
- “sample (S1-4)”in Line 104 of Page 3 in revised part means that four batches of Taxilli Herba samples from Morus alba were numbered S1-1, S1-2, S1-3, and S1-4. S1, S2, etc. in Table 3 had different meanings from S1-4, which represented the numbers of Taxilli Herba samples from 7 hosts.
Point 5: What do the authors mean exactly when they say "speculate"?
Reply 5: Thanks for your professional advice and we are so sorry to make it unclear.
On the one hand, the chemical composition was identified by comparison with the previously established chemical composition database, and verified with the retention time and mass spectrometry data of the standards. On the other hand, the identification of other unknown chemical composition was inferred based on the fragment information of MS/MS with the combination of SciFinder (https://scifinder.cas.org/), HMDB (https://hmdb.ca/), CNKI (https://kns.cnki.net/) and related literature. Therefore, we thought it is more rigorous to use speculate to describe unknown chemical composition.
Point 6: Section 2.3.1, L. 11: Fig. 3 should go in this line
Reply 6: Thanks for your professional comments and we are sorry to make it unclear.
Figure 3 was originally placed in Line 15 of Page 9. It might be difficult to understand because of the problem about the version. The picture below was the location of Figure 3 in my version.
Point 7: In section 2.3.1, I do not see all the sub-sections for the 33 flavonoids, namely: dihydroflavones, dihydrofalvonols, falvonols, isoflavones, flavones, flavones and other flavonoids.
Reply 7: Thank you for your sincere advice.
Line 17-33 of Page 10-11 were the summary of dihydroflavones, dihydrofalvonols, including 5 compounds.
Line 34-45 of Page 11 were the summary of flavanes, including 4 compounds.
Line 46-62 of Page 11 were the summary of flavones, including 5 compounds.
Line 63-82 of Page 11-12 were the summary of falvonols, including 15 compounds.
The remaining four flavonoids were briefly described in lines 83-86 including isoflavones and other flavonoids. The reason for not summarizing the cleavage behavior of these four compounds as described above is that one compound is not enough to represent the class of compounds, so we thought it is not appropriate to summarize the cleavage law of this type of compound.
Point 8: What do the authors mean exactly when they say "differential" throughout the manuscript?
Reply 8: Thanks for your professional advice. We are so sorry to make it unclear.
“differential” is an adjective and have the same meaning as different, which was used to describe chemical constituents. Differential chemical constituents of Taxilli Herba samples from different hosts were found based on the VIP>1 obtained from the PLS-DA model, and we used the T-test to verify whether the differential chemical constituents in multi-dimensional statistics had significant differences in one-dimensional statistics, where p<0.05 indicated significant differences.
We have checked and corrected other errors in the full text.
Once again, thank you very much for your comments and suggestions.
Round 3
Reviewer 4 Report
Some of the mistakes are still present in this latest version. The authors did not improve the description in section 2.3.1 regarding different types of flavonoids (already mentioned in the previous revision).
There are still many mistakes related to English style and orthography.